# Distinct genetic profiles influence body mass index between infancy and adolescence

Geng Wang ®[1] ✉, Samuel McEwan[1], Jian Zeng ®[1], Mekonnen Haile-Mariam[2,3], Loic Yengo ®[1], Michael E. Goddard[2,4], Kathryn E. Kemper[1,6] & Nicole M. Warrington ®[1,5,6] ✉

Body mass index (BMI) changes throughout life with age-varying genetic contributions. We use a random regression model to investigate the genetic contribution to BMI trajectories from ages one to 18 years in 6,291 ALSPAC participants with 65,930 repeated BMI measurements. Here we show the estimated SNP-based heritability of BMI at 9.5 years is 28.4% (SE = 4.8%), and 23.8% (SE = 4.2%) for rate of change in BMI from one to 18 years. The genetic correlations between early childhood and adolescence are low (genetic correlation between two and 17 years is 0.108 [SE = 0.146]). We find that the first principal component of the trajectory, explaining 89% of genetic variation, captures effects which increase in magnitude from early childhood to adolescence and then plateau. A second axis explaining 9% of the genetic variance has opposite effects on BMI between early and later ages. Our findings demonstrate the value of RRMs to reveal age-specific genetic influences on BMI across development.

Body mass index (BMI), defined as body weight (in kilograms) divided by the square of height (in metres), is a commonly used measure to estimate total body fat. Obesity, defined in adults by BMI exceeding 30 kg/m², poses a significant risk for the development of many diseases, particularly cardio-metabolic diseases[1]. One of the strongest predictors of obesity in adulthood is high BMI during childhood[2]. In the general population, BMI across childhood involves three distinct phases[3]. First, there's a rapid increase in BMI from birth to around 9 months, when children reach their adiposity peak (AP). Second, there is a rapid decline in BMI until around 5 or 6 years of age where children reach their adiposity rebound (AR), which is due to factors such as changes in body composition and increased physical activity. Finally, BMI gradually increases until early adulthood, reflecting ongoing body development through puberty. Increasing our understanding of the mechanisms influencing BMI across early life could lead to strategies to enable early prevention of obesity.

Considerable strides have been made in understanding the genetic underpinnings of BMI throughout the life course, where studies differ by both design and age of measurement. A key parameter is the heritability of BMI, i.e. the proportion of phenotypic variation attributable to additive genetic effects[4]. Twin studies, for example, estimate the heritability of BMI at age four to be around 40%[5] but this increases to around 80% for adolescents (10–19 years)[5–7] and adults[8]. A second design, based on genomic information, estimates the single nucleotide polymorphism (SNP)-based heritability, i.e. the proportion of variation in BMI tagged by common SNPs. SNP-based heritability estimates are generally about one-third to two-thirds of the twin-based estimates[4], where twin-based estimates are inflated by environmental or other confounders and SNP-based estimates fail to capture rare genetic effects[4]. For BMI, the SNP-based heritability is 20-40% during childhood (1 and 10 years of age)[6,9,10] and in adults[11]. BMI also exhibits genotype-by-age interactions. This means that the additive genetic effects differ by age and that the genetic correlation between ages is less than unity. For example, Silventoinen et al. pool data from 25 twin cohorts to show that the genetic correlation between BMI at age 4 and 18 years is 0.5 (95% CI 0.38–0.61) in males and 0.38

[1]Institute for Molecular Bioscience, The University of Queensland, Brisbane, QLD, Australia. [2]Agriculture Victoria, AgriBio, Centre for AgriBioscience, Bundoora, VIC, Australia. [3]School of Applied Systems Biology, La Trobe University, Bundoora, VIC, Australia. [4]Faculty of Science, The University of Melbourne, Parkville, VIC, Australia. [5]Frazer Institute, The University of Queensland, Brisbane, QLD, Australia. [6]These authors jointly supervised this work: Kathryn E. Kemper, Nicole M. Warrington. ✉e-mail: geng.wang@uq.edu.au; n.warrington@uq.edu.au

(95% CI 0.24 – 0.50) in females[12]. Moderate genetic correlations are also reported using SNP-based estimation by Helgeland et al.[10] between BMI at 5 years and in adults (0.43 SE = 0.041) and by Couto Alves et al. between BMI at the adiposity rebound and in adults (0.64 SE = 0.08)[13]. Thus although BMI is moderately to highly heritable in children and adults, the genetic factors influencing BMI are not consistent through time.

Helgeland et al. identify 46 genomic loci that are associated with BMI in at least one of twelve time points from birth to 8 years of age. Around half of these loci influence BMI during infancy but have no effect after the AR, including no effect on adult BMI. One locus following this pattern of association is *LEP/LEPR*, which has been associated with BMI across early life and at the adiposity peak in numerous studies[9,10,13]. Additionally, the genetic correlation between the BMI at the AP and adult BMI is lower than the correlation later in childhood (for example, $r_g$ = 0.26, SE = 0.07 between BMI at AP and adult BMI[13] and $r_g$ = 0.63, SE = 0.06 between BMI at 8 years and adult BMI[10]). The age-varying genetic effects at individual loci, along with the genetic correlations estimated to be less than one, indicate that there might be a unique genetic profile influencing BMI during the first few years of life that differs from the genetic profile affecting BMI in adulthood. However, all of these studies use cross-sectional data, which overlook individual-level changes and neglect to utilise the repeated measures data from longitudinal cohorts. Only a limited number of studies with smaller sample sizes have modelled growth patterns across time - these have shown promising and complementary findings to cross-sectional analyses[13–16].

One method for investigating genetic effects on traits over time using repeated measures data is the random regression model (RRM)[17–21]. The RRM offers flexibility in modelling the correlation structure of repeated measurements through the inclusion of random effects for each subject and accommodates missing or irregularly spaced data by leveraging all available repeated measurements per individual. Rather than imputing missing values, the method uses a mixed-effects framework to model population (fixed effects) and individual-specific effects (random effects) as a continuous function of time (e.g. age), thus estimating the population average trajectory as well as random coefficients to describe each individual's trajectory. The approach allows each participant to contribute to the estimation of longitudinal genetic parameters, even with incomplete follow-up data. Random effects can be partitioned into additive genetic and individual-specific effects, where the additive genetic effect can be estimated using a variance-covariance structure defined by a genetic relationship matrix constructed with common SNPs[22]. In contrast to previous studies using cross-sectional data[5,10,12], the RRM increases power to identify patterns of genetic variation by considering the effect of all SNPs simultaneously using all available repeated phenotype measurements. The RRM also reduces the number of parameters estimated from the data compared to previous twin studies which use a series of pairwise comparisons across ages to estimate genetic correlations[12]. It is flexible in that genetic correlations can be estimated at any given pair of ages (within the age range of the data). Previously, RRM has been used in humans with cross-sectional data to examine genotype-by-age and genotype-by-environment interactions[23,24], and here we apply the approach to longitudinal data. Finally, we can use RRM output to identify SNPs associated with features of the trajectory and conduct further analyses such as linkage disequilibrium (LD) score regression (LDSC) and Mendelian randomisation.

The aim of the current study is to use repeated measures data from a large birth cohort, the Avon Longitudinal Study of Parents and Children (ALSPAC) cohort, to investigate the genetic profile of BMI across early life and provide insights into whether it differs from BMI in adulthood. We estimate the SNP-based heritability and genetic correlations within the age range of one to 18 years, identify patterns of genetic variation in BMI, explore the influence of an adult BMI polygenic score on the genetic variance of childhood BMI, estimate the effect of individual SNPs on features of the BMI trajectory and estimate the genetic correlation between those features and adult traits and diseases.

## Results

A RRM with polynomials of age was used to model the additive genetic and unique individual differences of the BMI trajectory from 1 to 18 years using 65,930 BMI measurements from 6291 genotyped ALSPAC participants using ASReml-SA 4.2.1 (See 'Methods'). We estimated the SNP-based heritability and genetic correlations of BMI across early life.

Model comparisons indicated that the model with a quadratic polynomial for the additive genetic component ($k_g$ = 3), cubic polynomial for the random individual-specific effects ($k_i$ = 4), and a cubic polynomial for each sex in the fixed effects to model the overall population BMI trajectory was the superior fit to the data (Supplementary Table 1). Estimates of the fixed effects are presented in Supplementary Table 2, and the estimated covariance matrices for the random effects ($\mathbf{K_g}$ and $\mathbf{K_i}$) and residual variance ($\sigma_e^2$) are presented in Supplementary Table 3. The estimated trajectories from the RRM approximate the BMI measurements well for 16 randomly selected individuals (Supplementary Figs. 1 and 2), indicating good model fit.

### Genetic variance and SNP-based heritability

Our analyses indicate that common genetic variants across the genome influence average BMI as well as the shape of BMI trajectory over time (additive genetic variance component for the intercept: $\mathbf{K}_{g1,1}$ = 0.0073, SE = 0.0013; linear slope: $\mathbf{K}_{g2,2}$ = 0.0017, SE = 0.0003; quadratic polynomial: $\mathbf{K}_{g3,3}$ = 0.0004, SE = 0.0001; Supplementary Table 3). We detected a positive genetic correlation between the intercept and the linear slope (0.682, SE = 0.072), and a negative genetic correlation between quadratic polynomial and both the intercept (−0.678, SE = 0.132) and the linear slope (−0.473, SE = 0.174). The estimated SNP-based heritability of the intercept (28.4%, SE = 4.8%), linear slope (23.8%, SE = 4.2%) and quadratic polynomial (9.8%, SE = 3.1%) were all different from zero ($P$ < 0.05).

We transformed the phenotypic, additive genetic and unique individual variances between one and 18 years of age back to the observed age-scale (Fig. 1). We found the total phenotypic variance in BMI and all its components increase with age. The SNP-based heritability was significantly greater than zero and ranged between 23 and 30% across all ages, which is consistent with the SNP-based heritability estimate for the intercept (i.e. average age 9.5 years, Fig. 1 and Supplementary Table 4).

### Phenotypic and genetic correlations

The genetic correlation ($r_g$) between BMI at two different ages decreases as the difference between the ages increases (Fig. 2, Supplementary Fig. 3, Supplementary Table 5). For example, the genetic correlation between 1 and 2 years of age was high ($r_g$ = 0.948, SE = 0.015), whereas the genetic correlation between 1 and 10 years was not significantly different from zero ($r_g$ = −0.009, SE = 0.142). We note that as the difference between ages increases, the confidence interval around the estimate of genetic correlation increases, which is also observed when using LDSC. In contrast, the phenotypic correlation between BMI at age one and the subsequent ages decays quicker but remains non-zero (Supplementary Fig. 4, Supplementary Table 6). For example, the phenotypic correlation between 1 and 2 years of age was 0.67 (SE = 0.01), whereas the phenotypic correlation between 1 and 10 years was 0.19 (SE = 0.01). This means the genetic correlations at nearby ages are relatively smooth (Fig. 2 and Supplementary Fig. 3) but there are sharper peaks in the phenotypic correlations (Supplementary Fig. 4).

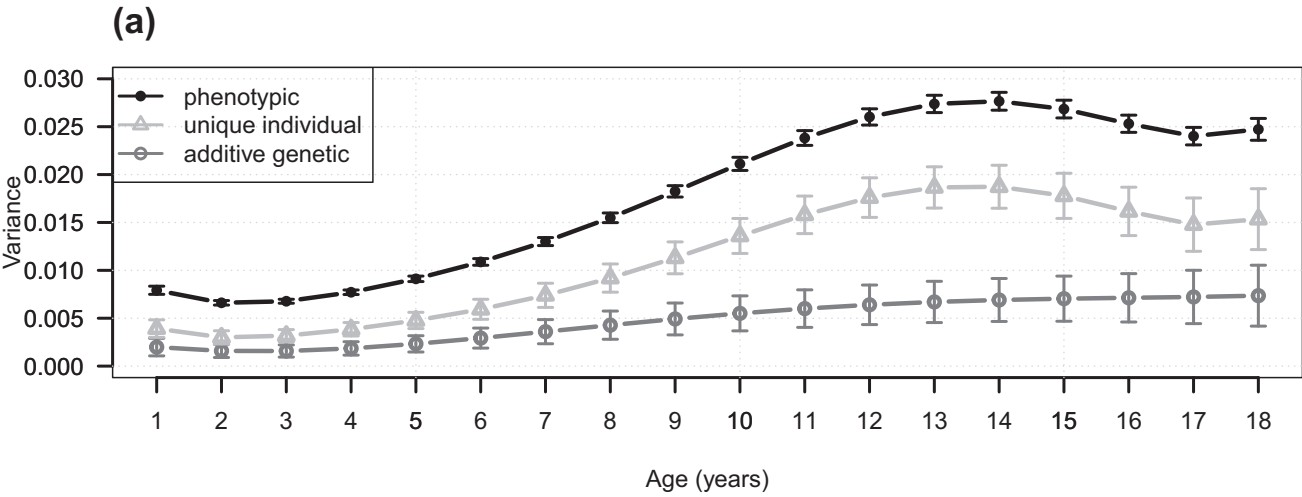

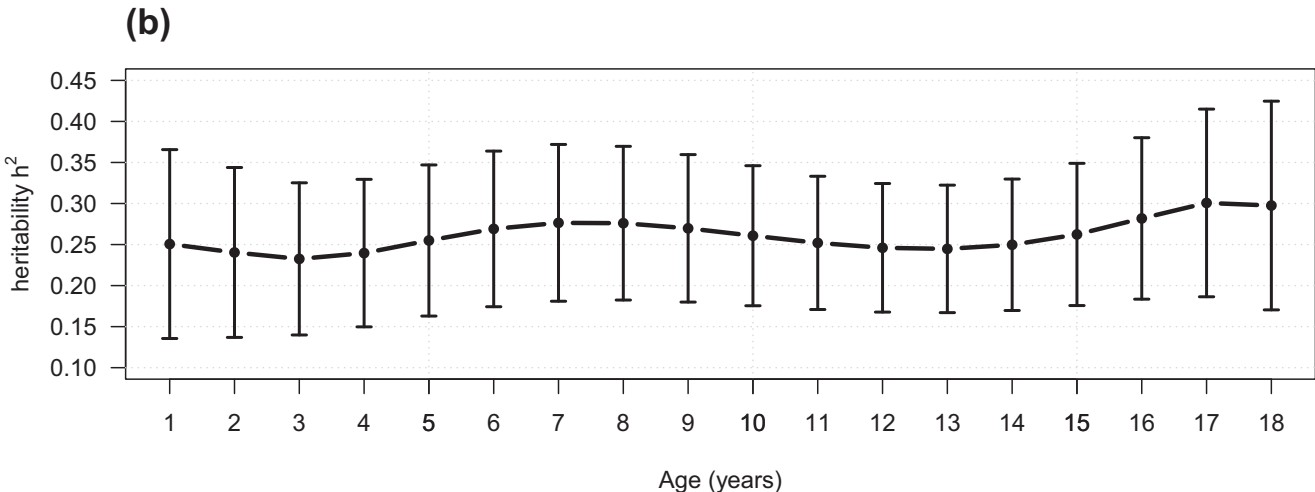

**Fig. 1 | Estimated variance components and SNP-based heritability from one to 18 years.** Estimated phenotypic, SNP-based additive genetic and unique individual variance components (**a**) and SNP-based heritability (**b**) from one to 18 years of age in the ALSPAC cohort. Points indicate the estimated phenotypic (solid point), SNP-based additive genetic (circle) and unique individual variance components (triangle) in (**a**) and the estimated SNP-based heritability in (**b**). The error bars indicate 95% confidence intervals. The sample size is 6291 ALSPAC participants with 65,930 repeated BMI measurements. Source data are provided as a Source Data file.

## Patterns of genetic variation

To identify age-varying genetic patterns of BMI, we conducted a principal component analysis (PCA) on $\mathbf{K_g}$ to obtain the three pairs of eigenvalues and their associated eigenvectors. Each principal component (PC) represents an independent (uncorrelated) axis of genetic variation. The first principal component (PC1) accounted for most of the variance in $\mathbf{K_g}$ (89% [95% CI 81%-96%]) while the second (PC2) explained ~9% (95% CI 4–20%) of the variance ($P = 0.02$ testing whether PC2 was different from zero). PC3 explained around 2% of the variance in $\mathbf{K_g}$ and was not significantly different from zero ($P = 0.10$) (Supplementary Fig. 5). To help with interpretation, we also transformed the eigenvectors into eigenfunctions, i.e. eigenvectors which are functions of time (See 'Methods'; Eq. 13). Figure 3 shows the eigenfunctions for PC1 and PC2 ($\psi_1$ and $\psi_2$) evaluated from 1 to 18 years of age, where the key feature of the eigenfunction is its relative change. Figure 3 shows that $\psi_1$ increases from zero over time until it reaches an approximate plateau at around 10 years of age. PC1 thus represents an axis of genetic variation with a consistent sign and increasing magnitude of effects on BMI across all ages. In contrast, $\psi_2$ crosses the x-axis at ~11.5 years, indicating that PC2 represents

genetic variation with opposing effects in infancy and adolescence (i.e. negative genetic covariance).

To investigate the PCs further, we generated polygenic score (PGS) for PC1 and PC2 and used them to categorise individuals into three clusters: those within one standard deviation of the mean PGS, and the remaining individuals with either high (greater than one standard deviation above the mean) or low (greater than one standard deviation below the mean) PGS scores. Supplementary Fig. 6 shows that individuals with high PGS for PC1 have a higher mean BMI, a steeper slope and a lack of adiposity rebound (usually around 6 years in children without obesity) in comparison to those with an average PGS. In contrast, individuals with low PGS for PC1 seem to follow more closely to a typical trajectory for BMI, but with a lower mean BMI and a slower increase in BMI after adiposity rebound. A similar investigation using PGS for PC2 (Supplementary Fig. 7) shows individuals with high PGS have a higher mean BMI at 1 year and a flatter trajectory across childhood resulting in a lower mean BMI at 18 years compared to the individuals with average and low PGS for PC2. This reflects the opposite genetic effects on BMI (relative to the mean) represented by PC2 in infancy and adolescence.

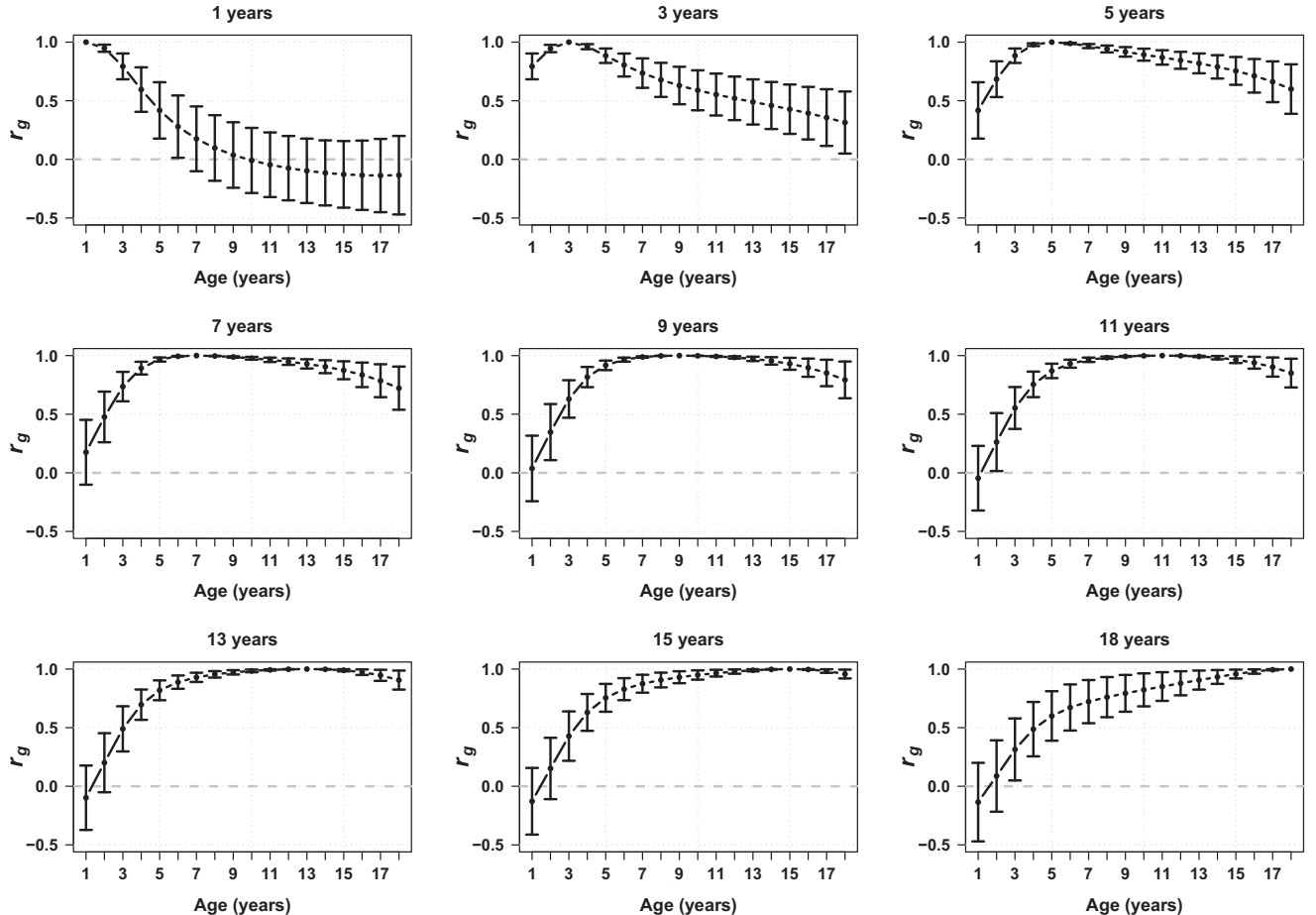

**Fig. 2 | Estimates of genetic correlation ($r_g$) of BMI between different ages from 1 to 18 years.** Points in each plot represents the estimated genetic correlations (y-axis) between a single age (described above the panel) and all other ages between 1 and 18 years (x-axis) in the ALSPAC cohort ($N = 6291$). Error bars represent the 95% confidence intervals. The grey horizontal line indicates zero genetic correlation. Source data are provided as a Source Data file.

## Adjustment of childhood BMI for adult BMI polygenic score (PGS)

We obtained a polygenic score (PGS) of adult BMI-associated loci and included it as an additional fixed effect in the RRM to investigate whether the genetic variance in childhood BMI could be explained by adult BMI loci (Supplementary Tables 7 and 8, Supplementary Fig. 8). We observed that the SNP-based heritability roughly halved after adjusting for the adult BMI PGS at age 18, reducing from 0.298 (SE = 0.065) in the unadjusted analysis to 0.145 (SE = 0.063) (Supplementary Fig. 9 and Supplementary Table 9). The 95% confidence intervals at all ages from 1 to 18 years differed from zero but the estimated heritability decreased with increasing age.

## Genome-wide association analysis and genetic correlations with adult disease

We fitted a reduced RRM, i.e. one excluding the additive genetic effects, to obtain individual (random effect) estimates of important parameters from our model for GWAS. These parameters include individual's intercept, linear and quadratic slope; as well as linear combinations of these parameters representing PC1 and PC2. We conducted a GWAS for these phenotypes and found the genomic inflation for all traits to be low (lambda GC = 1.015−1.034, Supplementary Fig. 10). We identified three loci associated with intercept ($P < 5 \times 10^{-8}$) in or near *ADCY3* (lead SNP: rs10203482-C, $\beta = 0.018$, SE = 0.0027, $P = 6.53 \times 10^{-11}$), *OLFM4* (lead SNP: rs4477562-T, $\beta = 0.023$, SE = 0.0041, $P = 3.98 \times 10^{-8}$) and *FTO* (lead SNP: rs55872725-T, $\beta = 0.016$, SE = 0.0028, $P = 5.31 \times 10^{-9}$, Supplementary Fig. 11). In

addition, SNPs within the *FTO* locus were associated with linear slope (lead SNP: rs55872725-T, $\beta = 0.0096$, SE = 0.0012, $P = 2.92 \times 10^{-15}$), quadratic polynomial (lead SNP: rs9972653-T, $\beta = -0.0046$, SE = 0.00074, $P = 4.38 \times 10^{-10}$) and PC1 (lead SNP: rs55872725-T, $\beta = 0.019$, SE = 0.0030, $P = 1.53 \times 10^{-10}$, Supplementary Fig. 11). SNPs within the *ADCY3* locus were also associated with PC1 (lead SNP: rs10203482-C, $\beta = 0.019$, SE = 0.0029, $P = 3.08 \times 10^{-10}$).

We used LDSC to estimate genetic correlations between the five derived BMI-trajectory phenotypes and a range of published GWAS summary statistics of 26 adult traits (Supplementary Data 1) and one childhood trait for validation. The SNP-based heritability estimates from LDSC were similar to those from the RRM (Supplementary Table 10), despite differences between the models. The LDSC has relatively large standard errors as it uses GWAS summary statistics rather than the individual level data. These large standard errors often resulted in unstable estimates for corresponding genetic correlations. However, we report them to urge others to replicate these findings. We found that the genetic correlations between the intercept, linear slope and quadratic polynomial were similar between LDSC and the RRM (Supplementary Table 11), and moderate genetic correlations between several adult traits and diseases and rate of change in BMI over childhood (linear slope; Fig. 4 and Supplementary Data 1); including triglycerides ($r_g = 0.317$, 95% CI [0.203, 0.431]), high-density lipoprotein (HDL, $r_g = -0.401$, 95% CI [−0.546, −0.256]), apolipoprotein A1 (ApoA1, $r_g = -0.306$, 95% CI [−0.434, −0.178]), HbA1c ($r_g = 0.339$, 95% CI [0.210, 0.468]), glucose ($r_g = 0.282$, 95% CI [0.147, 0.417]), risk of type 2 diabetes ($r_g = 0.578$, SE = 0.135) and risk of hypertension ($r_g = 0.235$,

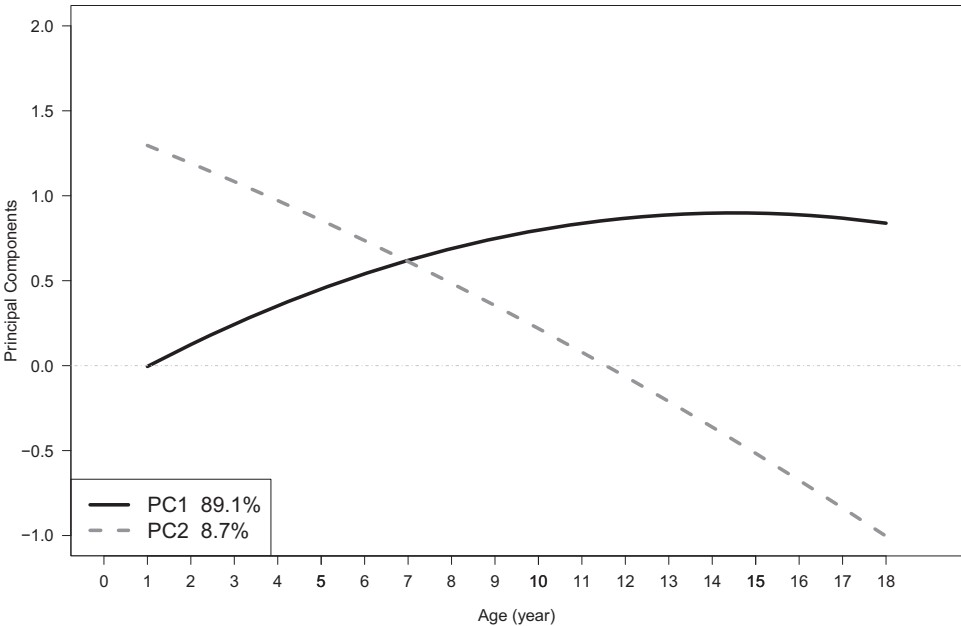

**Fig. 3 | Principal components 1 and 2, representing independent patterns of additive genetic variation on BMI, evaluated as functions of age from 1 to 18 years.** We conducted a principal component analysis on the genetic covariance matrix and this figure shows eigenfunctions representing the 1st and 2nd principal components (PC1 and PC2). The x-axis denotes age in years, while the y-axis represents the value of the eigenfunction. The black solid line represents PC1 and explains 89.1% of genetic variance. The dark grey dashed line represents PC2 which captures 8.7% of the genetic variance. Source data are provided as a Source Data file.

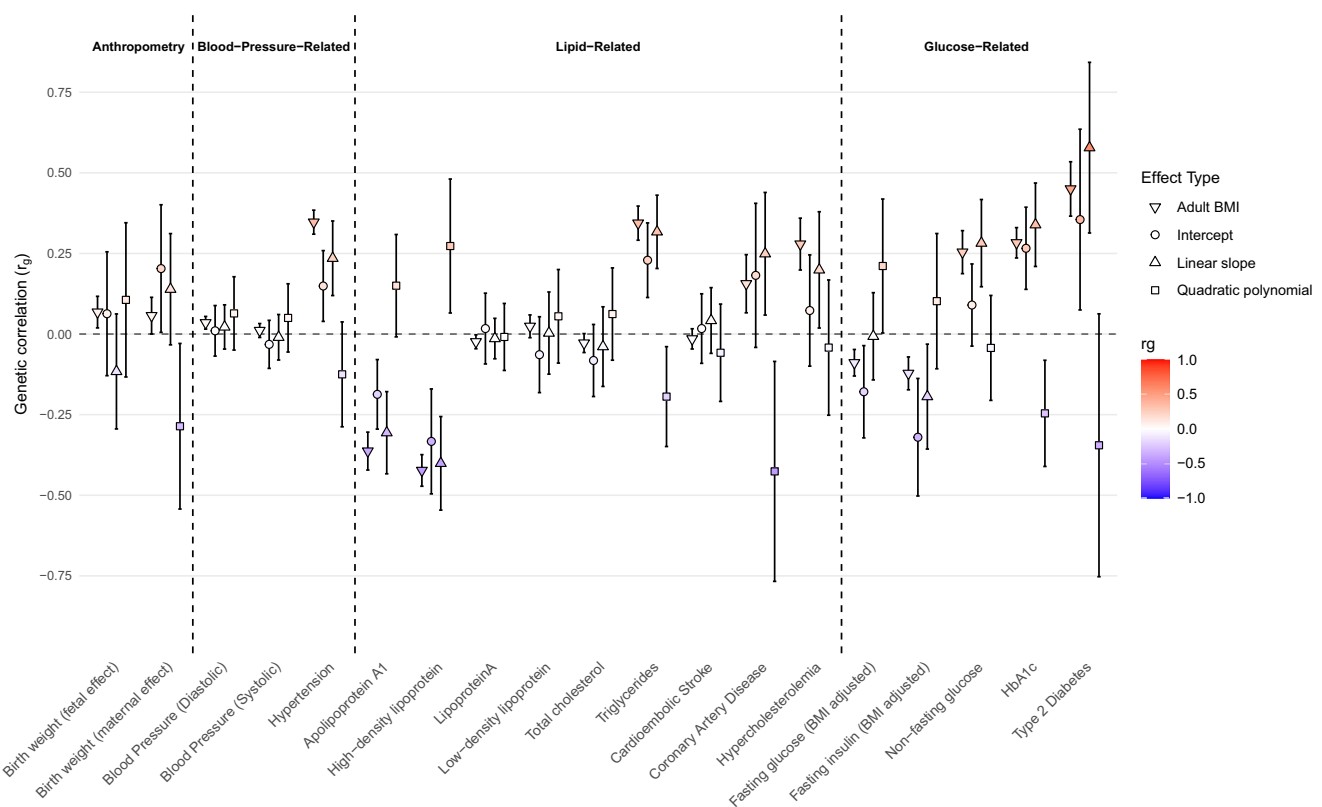

**Fig. 4 | Genome-wide genetic correlation between features of the BMI trajectory and a range of traits and diseases in later life.** Genetic correlation ($r_g$, shown in data points) between features of the BMI trajectory and the traits, and corresponding 95% CIs, were estimated using linkage disequilibrium score regression. $r_g$ between adult BMI and the selected traits is presented as a reference. The genetic correlation estimates are colour coded according to their intensity and direction (red, positive correlation; blue, negative correlation). See Supplementary Data 1 for the references for each of the traits displayed, as well as the genetic correlation results for other traits. Source data are provided as a Source Data file.

95% CI [0.119, 0.351]). The genetic correlations between the top two PCs and selected traits and diseases are presented in Supplementary Data 1 and Supplementary Fig. 12. As expected, some adult traits showed opposite genetic correlations with PC1 and PC2 (e.g. HDL, triglycerides, ApoA1, risk of hypertension and risk of type 2 diabetes), consistent with the opposing direction of their genetic effects at age 18 (Fig. 3). The results of validation and model checking are presented in Supplementary Note 1 (Supplementary Tables 12−16 and Supplementary Figs. 12−16)[10].

## Discussion

In the current study, we used a RRM to characterise the genetic profile of BMI from infancy to early adulthood using the ALSPAC cohort. We found significant additive genetic variation affecting the shape of the BMI trajectory from 1 to 18 years. In other words, there was genetic variation in the mean (or intercept, i.e $\hat{h}^2_{SNP}$ intercept = 28.4%, SE = 4.8%) of the BMI trajectory, as well as the parameters describing its shape (i.e. $\hat{h}^2_{SNP}$ linear slope = 23.8%, SE = 4.2%, $\hat{h}^2_{SNP}$ quadratic polynomial = 9.8%, SE = 3.1%). Since the interpretation of these shape parameters can be tricky, we transformed our estimates back onto the observed age-scale. There was a simultaneous increase in both additive genetic variance and unique individual variance over time, resulting in relatively constant SNP-based heritability ($\hat{h}^2_{SNP}$) across early life, ranging from 23 to 30%, and not significantly different from the heritability estimated for the intercept. The low genetic correlations found between BMI in infancy and later childhood, in contrast to the high genetic correlations at subsequent ages, indicate that there are different genetic profiles for BMI across the life-course. Additionally, the SNP-based heritability in early childhood was relatively unaffected by adjusting for an adult BMI PGS, whereas the SNP-based heritability in later childhood attenuated. PCs of the genetic variance-covariance matrix, $\mathbf{K_g}$, indicated that most of the genetic variation in BMI throughout childhood (PC1) acts consistently and progressively amplifies until about 10 years of age. Finally, we replicated previous findings that BMI at age 9.5 years (the intercept from our RRM) and change in BMI over childhood are associated with known adult BMI (*FTO* and *ADCY3*)[11] and childhood obesity/BMI (*OLFM4*)[25,26] loci. The association between variants in *FTO* and the intercept, linear slope and quadratic terms suggests that *FTO* variants influence mean BMI (around 9.5 years), rate of change and acceleration of BMI change during childhood, consistent with prior evidence of age-dependent genetic effects[25,27]. By contrast, variants in *ADCY3* are associated with only the intercept, implying a more constant influence on BMI across childhood without altering the rate of change, again consistent with previous studies[25]. These findings underscore the value of longitudinal GWAS for capturing dynamic genetic effects over development and suggest potential benefits for future meta-analysis of longitudinal GWAS efforts. We show that BMI at age 9.5 years and change in BMI over childhood are genetically correlated with a range of cardiometabolic traits in later life, including adult BMI, glucose-related traits, cholesterol and risk of hypertension and type 2 diabetes. While previous studies have established correlations between childhood BMI analysed cross-sectionally and cardio-metabolic risks in later life[10,26], the present analysis additionally examines genetic correlations with the rate of change in BMI across childhood.

Our study demonstrates that genetic variation contributes to childhood BMI trajectories. In clinical settings, paediatricians are primarily concerned with individuals who are underweight (often defined as being less than the 5th percentile on the Centers for Disease Control and Prevention (CDC) percentile charts[28]) or obese (often defined as being greater than the 95th percentile[28]). The CDC BMI-for-age percentile chart shows that the higher percentiles also have a greater rate of growth from 2 to 18 years of age. Here, we can ascribe some of the variability in growth curve percentiles to genetic factors. We also note a relatively strong genetic correlation between

the mean and linear rate of change (i.e. the genetic correlation between the intercept and linear slope in our RRM is 0.682 [SE = 0.072]), which suggests that the correlation between BMI at mean age (i.e. 9.5 years) and rate of change in BMI across childhood can be partly explained by genetics.

The SNP-based heritability on change in BMI from one to 18 years ($\hat{h}^2_{SNP}$ linear slope = 23.8%, SE = 4.2%), which has not previously been described, is substantially higher than that for change in BMI in adulthood ($\hat{h}^2_{SNP}$ linear slope = 1.98%)[29]. This indicates that the between individual rate of change in BMI across childhood has a genetic component, whereas rate of change in adulthood is predominantly driven by environmental factors. However, our observed SNP-based heritability estimates on mean BMI across childhood ($\hat{h}^2_{SNP}$ ~ 23−30%) were consistent with the SNP-based heritability for adult BMI[11] ($\hat{h}^2_{SNP}$ = 22.4% SE = 3.7%) and also the Norwegian Mother, Father and Child Cohort Study (MoBa) in infancy and early childhood[9,10] ($\hat{h}^2_{SNP}$ ~ 30−40%). In the CODATwins study[5], they observed an increase in the heritability from 4 years (42%, 95% CI 37 − 47% in boys; and 41%, 95% CI 35−46% in girls) to 19 years of age (75%, 95% CI 67−80% in boys; and 75%, 95% CI 67−82% in girls). Although we expect the heritability estimates from twin designs to be about double the SNP-based estimates[24], our data does not support an increase in heritability during childhood. This difference could arise from a relative increase in the importance of rare genetic variants influencing BMI, i.e. variants not captured by common SNPs, or other factors associated with twin designs[24]. We observed that the additive genetic variance captured by common SNPs increased throughout childhood from 0.0020 (SE = 0.00047) at year one to 0.0074 (SE = 0.0016) at 18 years. However, there was also a simultaneous increase in unique individual variance and, thus, no significant change in the SNP-based heritability over time.

Our results show decreasing age-to-age genetic correlations as the difference between ages increased from one to 18 years. This is consistent with previous studies including the CODATwins study[12], the MoBa study[10] and the Early Growth Genetics study[13] of BMI at the adiposity peak and adiposity rebound. The genetic correlation estimates from the RRM between 1 year and from 6 years onwards were lower than in the previous studies, but the standard errors were large. The large standard errors a likely due to the relatively limited amount of data between one and 2 years of age in ALSPAC and the exclusion of data prior to age one. These decreasing age-to-age genetic correlations suggest different genetic influences affecting BMI in infancy and early childhood compared to late childhood and adolescence. Our findings align with the transient genetic effects of SNPs detected in early life but not later life, as found in previous studies (e.g. *LEPR*[10,13]), which suggests that larger GWAS are needed to finely map the genetic profile of BMI during early life stages.

While genetic correlation focuses specifically on shared genetic influences between ages, phenotypic correlation captures overall associations between ages, regardless of their underlying causes. Previous studies investigating phenotypic risks for childhood obesity, such as Geserick et al.[30], have highlighted weight gain between 2 and 6 years as a key predictor of obesity in adolescence. Our study found that the phenotypic correlation between BMI at 18 years of age and all earlier ages progressively increased from $r_p$ = 0.16 (SE = 0.021) at age one to $r_p$ = 0.88 (SE = 0.003) at age 17, indicating that high BMI between ages 2 and 6 years might not be the only key time point for predicting obesity in adolescence. Interestingly, the phenotypic correlations were weaker than the (SNP-based) genetic correlations among nearby ages ($r_p$ = 0.67 [SE = 0.01], $r_g$ = 0.95 [SE = 0.02] between 1 and 2 years) but demonstrated greater strength in ages further apart ($r_p$ = 0.16 [SE = 0.02], $r_g$ = −0.14 [SE = 0.17] between 1 and 18 years). As the age difference narrows, the model predicts that the genetic correlation approaches 1.0 whereas the phenotypic correlation does not because all measurements are subject to random errors which are independent. The lower genetic correlation than phenotypic

correlation when the ages are far apart implies that genetic effects on BMI are more age specific than environmental effects on BMI.

The PCA of the genetic variance-covariance matrix allows us to explore the proposed model by Couto Alves et al. of genetic influences on childhood BMI[13]. This model, based on GWAS of key BMI trajectory features such as BMI at the adiposity peak and rebound, proposes two distinct biological factors underlying childhood BMI. The first acts primarily in infancy until roughly 8 years of age, and the second acts from birth with increasing strength until 18 years. Our results agree that the majority of genetic variance in childhood BMI (89%, 95% CI 81%-96%) is associated with factors which progressively amplify throughout childhood and act consistently from adolescence through to adulthood. These factors do not have a strong influence on BMI during infancy. Our results also indicate the second (orthogonal) axis of variation acts primarily during infancy with decreasing importance during childhood, and weak but opposing effects during adolescence. Therefore, by using PCA on the genetic variance captured in our model, we are able to provide stronger evidence for the proposed model of Couto Alves et al.[13] by defining two statistically independent axes of genetic variation influencing childhood BMI and determining the degree of variance associated with each axis.

The analysis adjusting for an adult BMI PGS aimed to (partially) account for genetic factors influencing adult BMI during childhood. Zheng et al. found the PGS explained 16% of the variance in adult BMI[31]. We observed a similar magnitude of attenuation in the estimated SNP-based heritability at 18 years old (by about 15%) after adjustment of the PGS, which is consistent with the adult PGS being an imperfect predictor of the genetic variance tagged by common SNP in adults. These results were also in line with the genetic correlation and PCA, whereby adjustment for the adult BMI PGS did not influence the variance components or SNP-based heritability during infancy (<3 years). This indicates that the genetic contribution to BMI during childhood to adolescence is shared with that of mid-to-late adulthood and highlights the unique genetic underpinnings of BMI during the infancy period.

While our findings are broadly consistent with previous studies of BMI genetics across development[9,10], the novelty of our study lies in the application of random regression modelling to estimate the genetic covariance structure ($K_g$) continuously across age. This approach offers flexibility in modelling trajectories using polynomials of age and provides insights into how genetics influences BMI change throughout early life. RRMs are computationally demanding, particularly for genome-wide analyses in large samples. In our implementation using ASReml, the model required at least 13.6 GB of primary workspace, with peak memory usage at 8.36 GB for the ALSPAC data. Despite this, scalability is feasible. As shown in Kemper et al.[32] a two-step approach can be employed by direct calculation of individual trait mean and rate of change, and then using a bivariate GREML approach to estimate the genetic (co)variance components. For our GWAS, we used a similar approach to Burrows et al.[15], whereby we used the individual random effects as phenotypes for the intercept, linear slope, quadratic polynomial to identified several known BMI loci. Although we were underpowered to identify novel loci in this study due to the limited sample size, this strategy makes it possible to extend the random effects modelling framework to larger cohorts and enables future discovery of age-dependent genetic loci influencing BMI.

The strengths of the current study include the utilisation of RRM, a model rarely used in human genetics research, in combination with data from ALSPAC, a comprehensive long-term birth cohort. While the use of a continuous function for genetic variance components of the BMI trajectory estimates the global genetic effects on the change in BMI over childhood, it also enables inference of the parameters (e.g. SNP-based heritability or genetic correlation) at any age on the BMI trajectory. It also leveraged the large

number of repeated measurements per individual (an average of 8 BMI measures per individual), improving the precision of our estimates over a traditional cross-sectional approach. While model diagnostics were appropriate using this continuous function in the ALSPAC data, if systematic deviations are detected, then a different modelling approach may be required. For example, the polynomial function may fail to capture the change in slope around the adiposity rebound (i.e. it over-smooths the trajectory), or there might be different variance structures before and after the adiposity rebound, then a more complex spline function[15,25] or splitting the data into two periods[13,27] might be worthwhile. However, identifying the most appropriate age to add a knot point for the spline function or split the data is often not trivial. We also showed how a PCA of the genetic variance-covariance matrix could revealed patterns of genetic variation in BMI that change over time. For the first time, our approach validates the proposed model of childhood BMI by Couto-Alves and colleagues[13]. Similar models, including linear mixed models[13,15,25] and Super-Imposition by Translation And Rotation (SITAR)[14], have been previously used to estimate the effects of individual SNPs on features of the growth (BMI or height) trajectory, but modelling all SNPs simultaneously has not previously been considered. Finally, we used the random effects estimates for each individual to estimate SNP effects on the features of BMI growth, identifying three known BMI associated loci. This allowed us to investigate the genetic correlation between these features of BMI growth and a range of traits and diseases in adulthood. We have shown that BMI growth is associated with a range of cardiometabolic traits; however, given the genetic correlations with adult BMI are of similar magnitude, it is likely that adult BMI is on the causal pathway between childhood BMI growth and these adult traits. Therefore, further investigation into these genetic correlations using causal modelling is warranted.

There are several limitations in our study also. Firstly, our estimates of the variance components and SNP-based heritability might be imprecise due to sample size, in comparison to those by the larger MoBa[10] and CODATwins cohorts[5], and we were therefore unable to identify fluctuations in SNP-based heritability across childhood. However, RRM estimates have narrower confidence intervals for genetic variation than cross-sectional studies with the same cohort size and can estimate genetic variation at any time point, even without direct data collection. Secondly, the non-significant low or negative genetic correlation in BMI between ages further apart should be interpreted with caution, as genetic correlations are estimated with lower accuracy compared to phenotypic correlations, especially when the age intervals are large. Thirdly, although the overall fitting of the Legendre polynomial function is satisfactory, improvements in model fitting may be possible by exploring other functions for age within the random regression framework (e.g. splines). A comparison of polynomial and spline functions has previously been conducted[33], and splines have been used to successfully identify the association between individual SNPs and BMI trajectories[15,25]. Finally, although ALSPAC is one of the most comprehensive longitudinal datasets available for such analyses, it is not without limitations. Approximately half of the cohort have provided consent and been genotyped, limiting the sample size available for genetic analyses. This has limited our ability to perform sex-stratified analyses, even though adiposity traits have previously been shown to be sexually dimorphic (for example, Pulit et al. suggest that approximately one-third of all genome-wide associated signals for waist-to-hip ratio are sexually dimorphic[34]). There is a sparsity of data between ages 1 and 7 years, particularly outside the CIF subsample, where the BMI trajectory changes substantially. For example, of the 6291 ALSPAC participants included in the analysis, 463 (7%) have no BMI measurements between 1 and 7 years when the regular clinic follow-ups of all participants began, 931 (15%) have one measure and 2209 (35%) have two measures. Additionally, previous studies have

shown that ALSPAC participants are generally of higher socio-economic position[35,36] and drop out from the study is associated with BMI[37,38]. This indicates that the missing at random assumption of the RRM might not hold. Therefore, future replication of these findings in more diverse cohorts with dense longitudinal data in early childhood (particularly before the age of seven, where data is sparse in ALSPAC) and different drop-out mechanisms is warranted.

In summary, the current study shows that there is a strong genetic drive regulating BMI during childhood and adolescence. Investigating the genetics of BMI during infancy is likely to identify genetic variants that differ from loci associated with adult BMI. Our findings provide justification for exploring the genetics of childhood BMI (e.g. through GWAS), as it is likely to discover genetic variants distinct from those loci associated with adult BMI. It also highlights the presence of age-varying genetic effects on BMI, emphasising the need to consider these factors when studying the genetics of childhood BMI, its relationship to adult BMI and future disease risk.

## Methods

Ethical approval for the study was obtained from the ALSPAC Ethics and Law Committee and the Local Research Ethics Committees. Informed consent for the use of data collected via questionnaires and clinics was obtained from participants or their mothers following the recommendations of the ALSPAC Ethics and Law Committee at the time. Data access of the ALSPAC cohort can be applied for by submitting a request to the study's Data Access Committee. Requirements for data access are described at http://www.bristol.ac.uk/alspac/. This project received ethical approval from the Institutional Human Research Ethics Committee, University of Queensland (Approval Number 2019002705).

### Study sample

ALSPAC is a population-based, prospective birth cohort conducted in region of Avon in the United Kingdom[35,36]. Pregnant women resident in Avon with expected dates of delivery between 1st April 1991 and 31st December 1992 were invited to take part in the study. The initial number of pregnancies enroled was 14,541, with 13,988 children who were alive at 1 year of age. When the oldest children were ~7 years of age, an attempt was made to bolster the initial sample with eligible families, resulting in an additional 913 children joining the ALSPAC cohort. The total sample size for analyses using any data collected after the age of seven is therefore 15,447 pregnancies, resulting in 15,658 fetuses. Of these 14,901 children were alive at 1 year of age. ALSPAC collected extensive health-related information from the mothers, fathers and their children at regular intervals from birth to early adulthood and blood samples from 9115 children were collected at various follow-ups for genotyping. Using information supplied by ALSPAC, we restricted our analysis to singleton births and individuals of European ancestry (see details below)[35,36]. The study website http://www.bristol.ac.uk/alspac/researchers/our-data/) contains details of all the data that is available through a fully searchable data dictionary and variable search tool[39].

### Genetic data

Genotyping, quality control (QC) and imputation procedures were performed centrally by ALSPAC and described in detail elsewhere[35,36]. In brief, the children were genotyped using the Illumina Human-Hap550 quad genome-wide microarray and imputed to the 1000 Genomes reference panel (Version 1, Phase 3, Dec 2013 Release, using haplotypes from all populations)[40] using IMPUTE2 (v2.2.2)[41,42]. QC measures applied centrally by ALSPAC included checks for gender mismatches, heterozygosity levels, missingness rates and Hardy-Weinberg equilibrium. Families who withdrew from the study were removed. Population stratification was evaluated using multidimensional scaling analysis, which was then compared to HapMap II

(release 22) reference populations[43], including individuals of European descent (CEU), Han Chinese, Japanese and Yoruba. The study removed all participants with non-European ancestry centrally. After QC and retaining only individuals of European ancestry from singleton births, there were 8635 genotyped children with 26,048,419 SNPs. More details of centrally-performed QC are provided at the following webpage: https://proposals.epi.bristol.ac.uk/alspac_omics_data_catalogue. html#.

We applied further QC steps to remove SNPs that displayed more than 5% missingness, a Hardy-Weinberg equilibrium $P$ value of less than $10^{-6}$, imputation quality INFO score less than 0.8, and minor allele frequency of less than 1%. A total of 6,284,594 SNPs were retained. Using the cleaned genotype data, we generated a genomic relationship matrix (GRM) using GCTA (version 1.94.1)[22]. Individuals with a relatedness coefficient of greater than 0.05 were identified and one member of every related pair randomly removed using the '--grm-cutoff 0.05' option in GCTA. This resulted in a set of 7791 unrelated individuals of European ancestry for further analyses. The top ten PCs were generated using the '--pca' option in GCTA.

### BMI measurements

The data collected in ALSPAC included height and weight at up to 32 follow-ups before children reached adulthood, including parent and child completed questionnaires, nurse reports from routine health care visits and clinic attendance.

We used the growthcleanr package[44] in R[45] to compare each height and weight measurement with a weighted moving average of the individual's other measurements to identify biologically implausible values in height and weight. Further details about the package are described elsewhere (https://carriedaymont.github.io/growthcleanr/articles/output.html)[44]. This flagged 5480 measurements (weight or height [2.1% of the total number of measurements], Supplementary Data 2), from 3667 individuals, as outliers and we removed them from subsequent analyses. Then, we only retained individuals of European ancestry with genotype information (see *Genetic Data*). Modelling BMI longitudinally from birth throughout childhood is challenging as the two inflection points (adiposity peak and rebound) make it difficult to find an appropriate function of age that accurately models the steep rise in BMI after birth then the more gradual changes through the adiposity rebound and puberty[33]. Therefore, we only used data collected between 1 and 18 years in our RRM. Firstly, we excluded data before 1 year so most individuals will have had their adiposity peak. Secondly, we excluded data after 18 years where BMI plateaus in most individuals. Finally, we only included individuals with at least four measurements, ensuring there is adequate data per individual to fit a cubic polynomial. The final dataset consisted of 65,930 BMI measurements from 6,291 unrelated individuals (Supplementary Table 17, an average of 10.5 (SD = 3.8) measures per individual). All BMI phenotypes were natural log-transformed for analyses due to skewed distribution of BMI. All data cleaning procedures were performed in R (version 4.2.1).

In this manuscript, we defined infancy as from 28 days (from 1 year in the analysed data) to 23 months, early childhood ranges from 2 to 4 years, late childhood covers ages 5–11 years, and adolescence from 12 to 18 years[46].

### Statistical model

The BMI trajectory for each person can be conceptualised as a combination of an average BMI curve that is common to everybody and described by the fixed effects of the RRM, and a departure from this average curve for the individual described by the random genetic effects and the random effects specific to that person. We used Legendre polynomials of standardised age in the RRM[20] to model the shape of the BMI curve over childhood. Legendre polynomials are a

system of orthogonal polynomials over the interval $[-1, 1]$ that reduce the correlation among regression coefficients, making them computationally efficient for longitudinal modelling. To generate Legendre polynomials, we first needed to standardise age ($m$) to the interval $[-1, 1]$ using the following equation:

$$m_{ij} = -1 + 2\left(\frac{age_{ij} - \min(age)}{\max(age) - \min(age)}\right) \quad (1)$$

where $age_{ij}$ is the age of individual $i$ at the measurement $j$, $\min(age)$ is the youngest age (i.e 1 year) and $\max(age)$ is the oldest age (i.e. 18 years) in the data.

We subsequently used the standardised age, $m$, to generate a matrix of Legendre polynomials evaluated at specified ages, $\mathbf{\Phi}$:

$$\mathbf{\Phi} = \mathbf{M\Lambda} \quad (2)$$

where $\mathbf{\Phi}$ is a matrix of dimension $t \times k$, where $t$ is the standardised age and $k$ is the degree of the polynomial plus one (in our case, $k = 4$; for example, coefficients for the overall mean [intercept], linear, quadratic and cubic polynomials of standardised age). $\mathbf{M}$ is a matrix of order $t \times k$ containing the standardised age values in their $m^k$ form (i.e. $m^0$, $m^1$, $m^2$, $m^3$). $\mathbf{\Lambda}$ is a matrix of Legendre polynomial coefficients of order $k \times k$, taking the form[47]:

$$\mathbf{\Lambda} = \begin{bmatrix} 0.7071 & 0 & -0.7906 & 0 \\ 0 & 1.2247 & 0 & -2.8067 \\ 0 & 0 & 2.3717 & 0 \\ 0 & 0 & 0 & 4.6771 \end{bmatrix} \quad (3)$$

Note we used $\mathbf{\Phi}$ in two different contexts below. First, rows from $\mathbf{\Phi}$ were used in the design matrices of the RRM model when $\mathbf{M}$ contains the observed ages where BMI was collected. Second, we used regularly spaced ages along the interval $[-1, 1]$ for $\mathbf{M}$ when transforming variance components from the RRM back onto the observed age-scale.

The following RRM was fitted to the observed data:

$$\mathbf{y} = \mathbf{Xb} + \mathbf{Z_g g} + \mathbf{Z_i i} + \mathbf{e} \quad (4)$$

where $\mathbf{y}$ is a $n \times 1$ column vector of log-transformed BMI observations ($n$ is the total number of observations). $\mathbf{X}$ is a $n \times (k_f + c)$ design matrix for the fixed effects, where $k_f$ is the degree of the polynomial plus one in the fixed effects, and $c$ is the number of covariates. The fixed effects include overall mean (or intercept), linear, quadratic and cubic Legendre polynomials, and covariates: sex, interaction terms between sex and the Legendre polynomials, and measurement source (whether the BMI measure was collected via questionnaire or at the clinic [including both the nurse report and ALSPAC clinic visits]). $\mathbf{b}$ is a $(k_f + c) \times 1$ vector of estimated fixed regression coefficients. $\mathbf{Z_g}$ and $\mathbf{Z_i}$ are $n \times (k_g \times N)$ and $n \times (k_i \times N)$ design matrices, respectively, for the random effects, where $N$ is the number of individuals and $k_g$ and $k_i$ is the degree of the polynomial plus one in the random effects ($k_g$ and $k_i$ are less than or equal to $k_f$). We have specified two unique design matrices, $\mathbf{Z_g}$ and $\mathbf{Z_i}$, to allow the degree of the polynomial (and therefore the value of $k_g$ and $k_i$) to differ between the additive genetic and unique individual effects. $\mathbf{g}$ is a $(k_g \times N) \times 1$ vector of random additive genetic effects, and $\mathbf{i}$ is a $(k_i \times N) \times 1$ vector of random individual-specific effects (i.e., individual-specific effects on BMI not captured by SNP genotypes such as, maternal effects, environmental effects and genetic factors that are not tagged by common SNPs). Finally, $\mathbf{e}$ is a $n \times 1$ vector of residuals, with the assumption that the variance of the residual term remains constant over time.

The distribution of random effects, $\mathbf{g}$, $\mathbf{i}$ and $\mathbf{e}$, follow a normal distribution with mean zero and variance given by:

$$\begin{aligned} \text{var}(\mathbf{g}) &= \mathbf{A} \otimes \mathbf{K_g} \\ \text{var}(\mathbf{i}) &= \mathbf{I_N} \otimes \mathbf{K_i} \\ \text{var}(\mathbf{e}) &= \mathbf{I_n}\sigma_e^2 \end{aligned} \quad (5)$$

where $\mathbf{A}$ is the genomic relationship matrix constructed with SNP genotypes, as described above, $\otimes$ denotes the Kronecker product, $\mathbf{K_g}$ and $\mathbf{K_i}$ are the $k_g \times k_g$ and $k_i \times k_i$ variance-covariance matrices for the additive genetic and unique individual effects, respectively, $\mathbf{I_N}$ and $\mathbf{I_n}$ are identity matrices with appropriate order for the respective random effects, and $\sigma_e^2$ is the variance of residuals. The model was fit using ASReml-SA version 4.2.1 (https://vsni.co.uk/software/asreml), with the ASReml model specification file (.as file) provided in Supplementary Note 2.

### Model fitting

We began by fitting cubic Legendre polynomials as fixed effects, and for both the additive genetic and unique individual effects, and then reduced the order of Legendre polynomials in the two random effects components of the RRM. We examined model fit visually in a subset of randomly selected individuals (Supplementary Figs. 1 and 2) and inspected the estimated variance components with standard errors. We used the Akaike Information Criterion (AIC) and a likelihood-ratio test (LRT) to formally determine the appropriate degree of polynomial function for the random effects, ensuring an accurate fit to the data (Supplementary Table 1).

### Transformation of variance components from the polynomials to the observed scale of 1 to 18 years

The genetic variance estimated by the RRM, $\mathbf{K_g}$, is a $k_g \times k_g$ variance-covariance matrix where the elements relate to the genetic variance of the intercept (or mean), linear slope and quadratic terms (when $k_g = 3$). To aid in interpretation, these polynomial terms can be transformed back onto the original scale of age using $\mathbf{\Phi}$, a matrix of Legendre polynomial coefficients (Eq. 2). As noted above, we can use any arbitrary number of values along the interval $[-1, 1]$ for $\mathbf{M}$ when calculating $\mathbf{\Phi}$ during this transformation as the variance components from the RRM are defined on a continuous scale. Thus, the estimates of the additive genetic ($\widehat{\mathbf{V_g}}$) and unique individual ($\widehat{\mathbf{V_i}}$) variance-covariance matrices on the observed scale are:

$$\widehat{\mathbf{V_g}} = \mathbf{\Phi K_g \Phi'} \quad (6)$$

$$\widehat{\mathbf{V_i}} = \mathbf{\Phi K_i \Phi'} \quad (7)$$

where $\widehat{\mathbf{V_g}}$ and $\widehat{\mathbf{V_i}}$ are of order $t \times t$, and $t$ is the number of ages across the BMI trajectory of interest for evaluation. For simplicity, we chose 18 equally spaced values along $[-1, 1]$ for $\mathbf{M}$ corresponding to yearly ages of 1 to 18 years, and thus $t = 18$. The estimates for the phenotypic variance ($\widehat{\mathbf{V_P}}$), SNP-based heritability ($\widehat{h_{SNP}^2}$), and genetic and phenotypic correlation between time points $t_1$ and $t_2$ [$\widehat{r_g(t_1, t_2)}$ and $\widehat{r_p(t_1, t_2)}$, respectively] were calculated as:

$$\widehat{\mathbf{V_P}} = \widehat{\mathbf{V_g}} + \widehat{\mathbf{V_i}} + \widehat{\sigma_e^2} \quad (8)$$

$$\widehat{h_{SNP}^2} = \text{diag}(\widehat{\mathbf{V_g}}/\widehat{\mathbf{V_P}}), \quad (9)$$

$$\widehat{r_g(t_1, t_2)} = \widehat{\mathbf{V_{g(t_1, t_2)}}} / \sqrt{\widehat{\mathbf{V_{g(t_1, t_1)}}} \widehat{\mathbf{V_{g(t_2, t_2)}}}} \quad (10)$$

$$r_p\widehat{(t_1, t_2)} = \mathbf{V}_{\widehat{\mathbf{P}(t_1, t_2)}} / \sqrt{\mathbf{V}_{\widehat{\mathbf{P}(t_1, t_1)}} \mathbf{V}_{\widehat{\mathbf{P}(t_2, t_2)}}} \tag{11}$$

where $\mathbf{V}_{\widehat{\mathbf{g}(t_1, t_2)}}$ is the $(t_1, t_2)$ element from the estimated additive genetic variance-covariance matrix $\widehat{\mathbf{V}_\mathbf{g}}$ and $\mathbf{V}_{\widehat{\mathbf{P}(t_1, t_2)}}$ is the $(t_1, t_2)$ element from the estimated phenotypic variance-covariance matrix $\widehat{\mathbf{V}_\mathbf{P}}$. Standard errors for the variance components, SNP-based heritability and genetic correlations were calculated using a Taylor series expansion following Fischer et al.[48].

## Patterns of genetic variation

We use a PCA on the genetic covariance matrix, $\mathbf{K_g}$, to identify independent patterns of genetic variation across the BMI trajectory. Each principal component (PC) captures an orthogonal axis of genetic variation over time – for example, a set of genetic variants that have stable genetic effects on BMI across age. Eigenvalues from the PCA indicate how much of the variance associated with each PC. This decomposition provides insight into how genetic effects change over childhood. PCA performs the following decomposition:

$$\mathbf{K_g} = \mathbf{EDE}' \tag{12}$$

where the matrix $\mathbf{K_g}$ (Eq. 5) is partitioned into its eigenvalues $\mathbf{D}$ and eigenvectors $\mathbf{E}$, using the eigen() function in R. To investigate the PCs further, the PCs (or eigenvectors) were transformed into eigenfunctions of age using $\mathbf{\Lambda E}$, where $\mathbf{\Lambda}$ is a matrix of Legendre polynomials (given in Eq. 3), and each column of $\mathbf{\Lambda E}$ represents the coefficients for each eigenfunction ($\psi$). The eigenfunctions for PC1 ($\psi_1$) and PC2 ($\psi_2$) were given by:

$$\psi_1(t) = 0.77 + 0.43t - 0.36t^2$$
$$\psi_2(t) = 0.29 - 1.15t - 0.14t^2 \tag{13}$$

where $t$ is the standardised age.

We tested if the variance explained by each PC was significantly different from zero using a chi-squared test[20]. The coefficient matrix ($\mathbf{E}$) is restricted by setting $q$ testing eigenvalues in $\mathbf{D}$ to zero, resulting in $\mathbf{D}^*$ and $\mathbf{K_g^*} = \mathbf{ED^*E}'$. The approximation can then be tested using a $\chi^2$ test with $q(q+1)/2$ degrees of freedom as:

$$\chi 2 = (\mathbf{K_g} - \mathbf{K_g^*})' \mathbf{V}^{-1} (\mathbf{K_g} - \mathbf{K_g^*}) \tag{14}$$

where $\mathbf{V}^{-1}$ is the inverse of the sampling variance-covariance matrix for $\mathbf{K_g}$ from the fitted model (obtained from the .vvp file in ASReml). We constructed 95% confidence intervals for the eigenvalues using numerical simulation (see Supplementary Note 3).

To help with interpretation, we also transformed the eigenvectors into eigenfunctions, i.e. eigenvectors which are functions of time.

## Evaluating BMI trajectories using the PCs

To further investigate the PCs, we calculated a polygenic score (PGS) for each PC and evaluated the BMI trajectory for low (<1 SD from the mean), average (within 1 SD of mean) and high (>1 SD from the mean) PGS individuals. In this context, the PGS evaluates the sum of additive genetic effects tagged by common SNPs for the primary (PC1) or secondary (PC2) patterns of genetic variation. The PGSs for the PCs were calculated as:

$$\widehat{\mathbf{G}} = \widehat{\alpha}\mathbf{E} \tag{15}$$

where $\widehat{\mathbf{G}}$ is a $N \times k_g$ matrix of the polygenic scores of the $N$ individuals for PC $i$ ($i = 1$ or 2), $\widehat{\alpha}$ is a $N \times k_g$ matrix of estimated random regression coefficients for the Legendre polynomials for the additive genetic effects and $\mathbf{E}$ is the matrix of eigenvectors (Eq. 12).

All analyses and plotting were performed in R (version 4.2.1). Two-sided P values of fixed effects and random variances were calculated using the chi-squared distribution function with one degree of freedom (Supplementary Table 2 and 3).

## Adjusting for adult BMI PGS

Next, we explored whether the SNP-based heritability across childhood was explained by the genetic variants known to be associated with adult BMI. We downloaded the SNP effect size estimates on adult BMI for ~7 million common SNPs generated from a SBayesRC[31] analysis of GWAS data of adult BMI in unrelated individuals of European ancestry ($N = 347,800$, age ≥ 40 years) from the UK Biobank[49,50] (https://sbayes.pctgplots.cloud.edu.au/data/SBayesRC/share/v1.0/PGS/). We constructed the adult BMI PGS in our cohort using the quality-controlled genetic data (see *Genetic Data* section) and the --score sum option in Plink v 1.9[51], which creates a sum of SNPs weighted by the adult BMI effect size. Subsequently, we included the adult BMI PGS and its interaction with age (i.e. interaction terms between PGS and the linear, quadratic, and cubic Legendre polynomials) as covariates in the fixed effects part of our RRM. The ASReml model specification file (.as file) is provided in Supplementary Note 4.

## Genome-wide association analysis and genetic correlations with adult disease

We fit a reduced RRM in ASReml that included only individual-specific random effects (i.e. removing the random additive genetic effects) for each polynomial term (intercept, linear slope, and quadratic polynomial). This reduces potential inflation of GWAS test statistics stemming from the inclusion of a GRM in the derivation of the phenotype. We then used the individual polynomial terms (intercept, slope and quadratic slope) and individual estimates of PC1 and PC2 (i.e. linear combinations of polynomial terms using PC1 and PC2 coefficients as weights) as phenotypes for GWAS. We performed genome-wide association analyses on each of the five derived phenotypes separately using PLINK v2.0, adjusting for the top 10 genetic PCs to account for population stratification. Gene names were assigned based on the nearest protein-coding gene using RefSeq annotations via HaploReg v4.2 (https://pubs.broadinstitute.org/mammals/haploreg/haploreg.php).

We used LDSC[52,53] to estimate genetic correlations between the GWAS summary statistics for each of the polynomial terms (intercept, linear slope and quadratic polynomial) and PCs with publicly available GWAS summary statistics for a range of outcomes previously linked with childhood BMI. We also estimated the genetic correlation between adult BMI (using summary statistics from Yengo et al.[11]) and each of the outcomes to contrast with our estimates on childhood BMI traits. The list of outcomes and publicly available GWAS summary statistics are reported in Supplementary Data 1.

## Validation and model checking

We aimed to ensure our interpretation of the SNP-based heritability, genetic correlations and genetic profiles estimated using the RRM were consistent with other approaches.

First, we estimated SNP-based heritability and genetic correlations in GCTA (version 1.94.1)[22] using selected measurements of BMI at cross-sectional follow-ups (i.e. average ages are 0.8 (closest cross-sectional time point to 1 year available), 1.7, 3.7, 7.6, 10.7, 13.9, 15.5 and 17.5 years). Each analysis included only one measurement per individual, with sex and age at measurement as covariates. We mostly used the BMI measurements of the same individuals included in the RRM analyses but included some additional records from the Child health database 2 (mean age = 0.8 year, $N = 4799$) and Teen Focus 4 (mean age = 17 years, $N = 3373$), which were excluded from the RRM as they were outside 1–18-year age range.

Second, our RRM (Eq. 4) assumes that the variance of the residual term is constant over ages. However, the phenotypic variance of BMI

increases with age, and therefore the variance of the residual term may also vary with age. This may introduce bias into the estimation of both additive genetic and unique individual variances. To address this concern, we estimated different residual terms for each year in our model (i.e. we fit 17 residual terms in the random regression model). The ASReml model specification file (.as file) is provided in Supplementary Note 5.

Third, we performed PCA on the genetic correlation matrix of BMI obtained from the previously published COADTwins project[12], which was described in Supplementary Note 6.

### Reporting summary
Further information on research design is available in the Nature Portfolio Reporting Summary linked to this article.

## Data availability
The ALSPAC data are available under restricted access in accordance with the ALSPAC access policy. Full instructions for applying for data access can be found via https://www.bristol.ac.uk/alspac/researchers/access/. The GWAS summary statistics have been deposited on the Program in Complex Trait Genomics website (https://cnsgenomics.com/content/data) for public access. The summary results data generated in this study are provided in the Supplementary Information/Source Data file. Source data are provided with this paper.

## Code availability
The main analysis utilises published software (i.e. ASReml) and packages (e.g. 'growthcleanr' in R). The custom R and ASReml code used to perform the analyses in this study are available in the Supplementary Materials.

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

## Acknowledgements

This research has been conducted using the ALSPAC (Reference B4194). We are extremely grateful to all the families who took part in this study, the midwives for their help in recruiting them, and the whole ALSPAC team, which includes interviewers, computer and laboratory technicians, clerical workers, research scientists, volunteers, managers, receptionists and nurses. We would like to express our sincere gratitude to Dominic Waters (The University of New England, Armidale, Australia) for provision of code from Waters et al.[54] for calculating standard errors for the parameter estimates. This publication is the work of the authors, and N.M.W. will serve as the guarantor for the contents of this paper. G.W. and N.M.W. are funded by a National Health and Medical Research Council (Australia) Investigator grant (APP2008723). The contents are solely the responsibility of the authors and do not reflect the views of NHMRC. K.E.K. was supported by the Australian Research Council (grants FL180100072 and DP200100499). L.Y. is supported by the Australian Research Council (grant no. FT220100069) and the Snow Medical Research Foundation. J.Z. acknowledges funding from the Australian National Health and Medical Research Council (1177268). The UK Medical Research Council and Wellcome (Grant ref: 217065/Z/19/Z) and the University of Bristol provide core support for ALSPAC. GWAS data in ALSPAC was generated by Sample Logistics and Genotyping Facilities at Wellcome Sanger Institute and LabCorp (Laboratory Corporation of America) using support from 23andMe. A comprehensive list of funding provided to ALSPAC from grants is available on the ALSPAC website (http://www.bristol.ac.uk/alspac/external/documents/grant-acknowledgements.pdf). This research was specifically funded by Wellcome Trust and MRC (core), 076467/Z/05/Z.

## Author contributions

G.W. designed the study, conducted the statistical analysis, interpreted the results and drafted the manuscript. S.M., J.Z. and M.H.-M. contributed to the statistical analysis and reviewed the manuscript. L.Y. contributed to data acquisition and reviewed the manuscript. M.E.G. interpreted the results and reviewed the manuscript. K.E.K. and N.M.W. designed the study, interpreted the results and reviewed the manuscript.

## Competing interests

The authors declare no competing interests.
