## [Transparent Peer Review file · Nature Communications]

Distinct genetic profiles influence body mass index between infancy and adolescence

Corresponding Author: Dr Geng Wang

Version 0:

Reviewer comments:

Reviewer #1

(Remarks to the Author)

Wang and colleagues here describe a new approach to investigate the genetic contribution to child BMI from infancy to late adolescence. They use random regression to utilize the longitudinal design of the ALSPAC cohort that has repeated measures of height and weight in up to 6291 genotyped children.

They quantify the SNP-based heritability to between 23-30% across childhood, identify age-varying genetic patterns of BMI by pairwise genetic correlation analyses and perform a comparison with a classical cross-sectional analysis in the same sample. They conclude that there is a different genetic profile operating during infancy compared to later childhood and adolescence.

The results from this novel method are largely consistent with previous studies that also have shown that the genetic influences on BMI changes with time from early to later childhood and to adulthood. As the authors point out – this is important as this implies that studies of genetics of BMI in childhood are likely to discover variants that differ from loci associated with adult BMI.

However, there are also some concerns with the study. While models taking into account the availability of repeated measurements over time have several advantages compared to cross-sectional approaches, they can also be susceptible to model misspecifications, in particular when the number of observations per individual are limited and there are rapid changes in the phenotype and genotype correlations.

The apparently complete disconnection of infant BMI-genetics with later childhood BMI (from 7 years and onwards) seen with their method stands out as the most novel finding and seems at odds with both previous studies and their own cross-sectional analysis in the ALSPAC data. As detailed below, this raises some concerns about the performance of the longitudinal modelling for the earlier timepoints.

There are also some other suggestions that may help improve the paper.

Main concerns.

1. The model performance at the earlier timepoints seems problematic. Several issues should be addressed
 - a. The authors suggest (p19) that the genetic contributions to BMI in early childhood (up to age6) appear to be independent of genetic influences in later childhood. This is at odds with previous larger studies that they refer to elsewhere in the manuscript, f ex Alves et al and Helgeland et al that show that there is indeed a lower, but significant shared genetic architecture between BMI at around age 1 and both later childhood and adulthood. It is of course possible that some of this apparent discrepancy could be explained with the very large CIs of the point estimates, but it is highlighted as one of the key findings of the paper and thus should be scrutinized.
 - b. The authors own pairwise correlations between cross-sectional models in Alspac suppl Table 11 shows a picture that is more similar to previous studies with point estimates 0.58 (SE:0.15) and 0.68 (SE:0.15) to age 7 for age 0.8Y and 1.7 Y respectively, and continued relatively strong positive correlation estimates with measures later in life. Again, large SEs can blur the picture, but conclusions from these two analyses seem to be largely contradictory.
 - c. Model fitting at ages 1-5 years of age. If the phenotypic and genetic correlations change drastically from 1 to 6 years of age as most studies suggest, including the current one, any modelling of the growths curves during this time will rely on their ability to capture these changes. Thus, the number and timepoints of measurements are very important. Table 1 and suppl

Fig 1 show that there are indeed very few measurements during this time for 90% of the participants. The BMI measurements in Alspac come largely from two different approaches. Alspac has a CIF-subsample of appr 700 children with dense measurements from 12 to 61 months, but for the remainder the main data are collected from around age 0.8Y, 1.7Y, 3.7Y, and then from age 7Y and onwards. As the authors exclude any measurements before 1Y of age, >90% of the samples have only at best measures at 1.7 and 3.7 years to guide their models in this critical time window. Suppl Fig 1 gives a rather clear illustration of this problem. Thus it can be questioned if the results for age1-3 are robust.

d. It looks like the fit at later timepoints also may be suboptimal as many subjects lack measurements after age 15. However, it is not likely that the genetic architecture is changing as much here, and thus, interpolation of the growth curve would suffer less.

2. The abstract could be clearer on the most novel aspects of the study.

3. What is the gain and novelty in the current approach compared to other longitudinal models? There seems to be another approach to longitudinal modelling with the same PI on medRxiv (doi: 10.1101/2024.03.13.24304263). It would be interesting with a discussion on the similarities and differences with this approach as well as other approaches to longitudinal modelling of growth related traits such as the recent study from EGG (<https://doi.org/10.1186/s13059-023-03136-z>)

4. The sample size limits the power to detect differences, but it is indeed interesting to note that the SNP-based h2 point estimates are generally higher in the cross-sectional approach in the current study, but also in previous studies, than the longitudinal approach presented here. Hence, a more extensive discussion on pros and cons for longitudinal vs cross-sectional approaches under various scenarios would be interesting. Alspac, with its two different designs (CIF with high density and higher quality measurements vs a sparser coverage of the majority of samples) could serve as inspiration for how to design new studies.

5. Why did they exclude related individuals, mixed models should allow to account for that?

6. As the results to a large confirm results from other previous larger studies, the most novel aspect of the study might the method. It would be interesting to know whether the random effects model consume a lot of memory and if it might be possible to scala the model to bigger sample sizes and whether it can be extended to enable identification of genetic variants too.

Reviewer #2

(Remarks to the Author)

Key results:

The results indicate that there are different genetic profiles that influence BMI between infancy and adolescence. This could be expected by biological developmental stages and previous studies. However, compared to previous reports this uses large number of measurements in a relatively large sample size and full BMI trajectories instead of cross-sectional timepoints plus additional techniques to evaluate age-varying genetic patterns in the BMI development. The weakness maybe a lack of replication in other studies where similar data are available (although some analyses were validated using COADTwins project). The result section is detailed and describes the observations well although it is quite heavy to read. The whole range of different analytical methods were used to illustrate and validate the result and to reach the best outputs from the rich data – excellent work indeed. Figure 2 is interesting, and informative but the details are blurred with a very small text (same concerns Figure 4 as well), so could be improved. Maybe in Figure 3 eigenfunctions could be better or more explicitly explained in the Figure legend. Some parts of the results will be difficult to grasp if someone is not very familiar with the methodology. Overall, as authors acknowledge, increasing our understanding of the mechanisms influencing BMI across early life could lead to strategies to enable early prevention of obesity.

Validity:

SEE BELOW: The study is highly valid, interesting and well conducted. Overall, methods and data have been well presented and they seem valid. However, some methodologies could be explained more explicitly particularly for readers that are not familiar with the approaches (e.g. use of eigenvalues). I also would like to acknowledge a high expertise of the authors of this work.

Significance:

This study is important and interesting because it summarises large amount of previously published information (good introduction + discussion) and shows how study design and approach itself has impact on suggested heritability estimates. Maybe it would be useful to strengthen the text in terms of that genetic correlations are very much dependent on which ages or growth patterns we are comparing. Information is there but it could possibly be spelled out or summarised better. The aspect and role of confounding is, however, discussed in relation to different types of studies (twin-studies, i.e. purely observational studies compared to those that use genetic information as well). The present and previous studies that have aimed to distil that how different genetic factors have impact on different growth stages are important as they reveal also biological pathways that are important for development of body mass. These data could then possibly be used in the future when designing prevention strategies that also integrate genetic information. Overall, we can say that genetic contribution to the development of obesity is substantial and matter for how environmental factors such as nutrition or exercises may impact fat accumulation. The earlier discovery of LEP/LEPR locus which has been associated with BMI specifically across early life and at the adiposity peak is very important (as noted in the present study, too). The important outcomes from this study were also, as authors state, that genetic effects on BMI are more age specific than environmental effects, and that larger GWASs are needed to finely map the genetic profile of BMI during early stages. This directs the future works.

Data and methodology:

The team has used the data from the Avon Longitudinal Study of Parents and Children (ALSPAC) cohort. Random

regression modelling was used to estimate the genetic covariance matrix of BMI trajectories from ages one to 18 years with 65,930 repeated BMI measurements from 6,291 children. Cut-off age 1 year was used due to the selected modelling option but justification could be better explained (basically this avoids adiposity peak in infancy in the most of the children, but reading the text it shows that measured under one year, ~0'8y, were also used in SNP based heritability cross-sectional analyses – which is fine, i.e. only measurements over 1 y of age were used in main longitudinal analyses). The data are unique and well-documented. The number of children is based on the availability of growth and genetic data which means that less than half of ALSPAC data was used in these analyses. Table 1 also shows the multiple sources of the data which may increase variability or inconsistency? This arises the question that how representative these data are compared to the original full data with same ethnic origin as used here (this information could be added into the supplementary information for example or at least discussed shortly). However, the decrease in the availability of the data is very common and practically unavoidable e.g. due to restricted resources and losses-to-follow up or withdrawal of the consent to use the data (as seemed to happen in ALSPAC as well). In spite of some caveats the study is highly valid, interesting and well conducted. Overall, methods and data have been well presented including inclusion and exclusion criteria that were multiple and explain to some extent the decrease in the sample size. Table 1 generally also shows the complexity in this kind of longitudinal data collections and formation of a final sample for the analyses. The information on genotyping is given at a detailed enough level.

Analytical approach:

The aims of the study and analytical approaches are generally well and clearly presented. As authors present one potential method for investigating genetic effects on traits over time using repeated measures data is the random regression model (RRM). These methods as such are not that novel or newly developed but are appropriate for these purposes because the model offers flexibility in modelling the correlation structure of repeated measurements through the inclusion of random effects for each subject. Authors write that RMM accommodates missing or incomplete data but it is not clear to me how it works in this respect. Does it impute automatically? The “power” of RMM over other methods is described and justified for the present work. Statistical analysis part is details with model descriptions and fitting (this part could be reviewed by specialised biostatistician) but as noted above the readers who are not familiar with all the methods would appreciate further overall description (e.g. in the main document how Eigenvalue decomposition works in spite of that in the supplementary note there is a more detailed description of this). In contrast to previous studies using cross-sectional data, the RMM increases the power to identify patterns of genetic variation by considering the effect of all SNPs simultaneously and it does not attempt to identify individual SNPs associated with a particular time-point or feature of the trajectory. However, these different approaches including time-point or feature of trajectory analyses are valuable and complimentary as they give different information for example in terms of potential preventative approaches (periods during the life-course). This aspect could be acknowledged. This study is also a nice validation and continuum for the study of Couto-Alves et al that was a very first study, as far as we know, to explore genetics of growth patterns and differences between developmental stages related to BMI growth (age-dependent height growth velocity analyses preceded this actually).

Suggested improvements:

In the text above I have suggested improvements of clarifications. Overall, this work is nicely presented and valuable. Replication would be an advantage.

Clarity and context:

Commented already above. Discussion is also well written, informative and includes key aspects to address. This study falls very well into the context of current urgent health matters that we will need to tackle from early childhood onwards.

References:

It looks to me that the main key references are included. Maybe one that examined genetics of PHV between infancy and adolescence would be useful to mention as it is as far as we know the first of its kind where research team tried to understand genetic differences between developmental stages. The paper shows that nicely.

Reviewer #3

(Remarks to the Author)

Wang et al. present a well-motivated, well-designed, and thorough manuscript describing the genetic contribution to body mass index (BMI) trajectories through early life. They utilise random regression modelling applied to repeated measures data from the ALSPAC cohort to estimate the SNP-based heritability of, and genetic correlations between, BMI across ages 1-18 years. The ability to model longitudinal data (with missingness) and simultaneously incorporate multiple common SNPs (through a genetic relatedness matrix) empowers the authors to infer the evolving genetic profile of BMI trajectories. They confirm previous findings of the different genetic components of infant, childhood, and adolescent BMI, with adult BMI genetics explaining more of the latter than the former. While methodologically advanced, I think the manuscript lacks new results, so I have made some suggestions below that may improve this.

Major Comments:

1. Page 5, first paragraph: “In contrast to previous studies... and it does not attempt to identify individual SNPs associated with a particular time-point or feature of the trajectory.” Is this meant to signify a strength of the work? I think it is a drawback

that the authors cannot identify putatively causal SNPs with this method and should be included in the limitations.

2. Page 7, last paragraph: It would be useful to know the reasons why outliers were removed (provided by the growthcleanr package), and the number of removals in different data sources. This could help inform future studies that are looking to aggregate data across multiple sources on the reliability of different sources for childhood BMI (for example, clinical vs self-report by a parent).

3. What are the inflection points in Figure 1B? Do they represent any known phenomena?

4. It would be really interesting to know if eigenfunctions 1 and 2 (or the PGS for these) explain any proportion of variance in adult diseases, such as adult obesity, breast cancer, type 2 diabetes risk, etc. Genetic factors associated with childhood BMI are protective against some adult diseases and risk factors for others – I wonder if your analysis can separate out the effects attributable to different genetic profiles?

5. Page 26, first full paragraph: Could the authors estimate the heritability from RRM at the mean ages used in the cross-sectional GCTA analysis to enable direct comparison? And add a P-value for the difference between the estimates from their RRM and the cross-sectional GCTA estimates.

6. Eigenfunction 2 appears to be distinct between males and females in the CODAtwins, i.e. it decreases monotonically in males but appears to go through a peak during AR and then declines in females. Could the authors check that this is also the case in ALSPAC? What are some possible reasons for this?

7. More broadly related to the point (6) above, it might be worth doing all the presented analyses separately in males and females, given the authors have noted that there may be different age-age genetic correlations in males and females in the Introduction (page 4, citation to Silventoinen et al.)

Minor Comments:

1. The Introduction is unnecessarily lengthy. For example, the paragraph on adult BMI heritability (page 3, first full paragraph) is not needed given the focus of the manuscript is on childhood BMI.

2. Page 3, last paragraph: In the sentence “For example, twin studies estimate a heritability of around 40% at age four, presumably where there is a large shared environmental component to BMI”, it is not clear to me what is meant by “where” and “shared environmental component”.

3. Page 4, first paragraph: Correlations of between 0.4 to 0.6 are described as “low”. I would argue that these are not low given the range of possible correlation values is only $[-1, 1]$ – perhaps use “moderate” instead?

4. Page 14, sentence above equation 13: I don’t understand the term in parentheses “(e.g. eigenvalue 3)” in relation to the rest of the sentence, as it currently seems as though q represent the number of eigenvalues (i.e. 1, 2, or 3) set to 0, rather than just one eigenvalue (i.e. the q th eigenvalue), being set to 0.

5. Page 21, sentence below Equation 16: “... we can see a relatively large [absolute] weight of the intercept...” – Please insert the word absolute as specified.

6. Figure 4A and Supp. Figure 6: Can the authors please add confidence intervals around the mean profiles?

7. Page 17, first paragraph of Results: Could the authors speculate on why the individual-specific effects require a cubic term while the population-level effects are quadratic? Has this been observed before in models of childhood BMI?

8. Supp. Table 7: It would be useful to present the RRM fixed effects from the model un-adjusted for adult BMI PGS side-by-side with the adjusted coefficients to enable comparison.

9. Page 26, first full paragraph: Could the authors plot the age-to-age genetic correlation patterns from GCTA cross-sectional analysis, the same way they have done from the RRM analysis, and present them together to enable direct comparison?

10. Page 26, last line in the first full paragraph: “... which was not different to the RRM..” – Please provide a P-value for the difference.

Reviewer #4

(Remarks to the Author)

Version 1:

Reviewer comments:

Reviewer #1

(Remarks to the Author)

I find the manuscript now much improved, and the addition of a GWAS-discovery really shows the applicability of the approach. The study is clearly an important and interesting contribution to the field.

I have only a few remaining questions.

1) Regarding the performance of the model during the earlier time-points. I appreciate the thorough answer and the changes to the manuscripts which I think have improved the paper. I still think the authors could help the readers and the design of future studies if they discuss a bit more the implications of their findings for the longitudinal modelling based approaches of BMI: In particular the implications based on the relatively rapid changes in genetic influences/correlations that seem to occur from 1 to 5 years of age and how it may affect the results.

If indeed there is a low genetic correlation, then, it must potentially be a concern when there is a non-random and quite dramatic lack of measurements for the earlier time-points. Based on the sFig1 plots, that shows 16 random individuals, there

are at least five individuals (inds 6,7, 9,11,and 14) that do not have a single measurement before the typical adiposity rebound at age 5, and four (2,4,10 and 12) that have only one measurement. I think it would be appropriate to be clearer about this in the limitations and possibilities section – as it might contribute to more noise (and even bias) when modelling these early life genetic influences – and may explain

1. The apparent lack of signal for LEPR and other AP-centred signals, but stronger performance for ADCY3 and FTO.
2. The lower heritability and correlation with later time-points compared to age-stratified approaches (where no modelling is done).

This raises the possibility to speculate a bit on how future studies could be targeted to seek to optimize the trade-off between retaining sample size and avoiding diluting the signals or even introduce bias to the analysis (for ALSPAC the performance at age 1-3 will be more dependent on the impact of later time-points – thus AP-centred signals will be diluted). What are the pros and cons for modelling the entire childhood trajectory, rather than splitting it in two: one pre- and one post-AR if they are indeed relatively un-correlated? A bit speculation here would be interesting to read.

A few additional comments:

- 2) Genetic correlations page 7: Please comment, in the following sentence, that these estimates between more distant time-points are surrounded by very huge confidence intervals.

“For example, the genetic correlation between one and two years of age was high ($r_g = 0.948$, $SE = 0.015$), whereas the genetic correlation between one and 10 years was not significantly different from zero ($r_g = -0.009$, $SE = 0.142$).”

- 3) Fig9 E. Please adapt the y-axis to allow the red line to be seen to avoid misinterpretation of genome wide sign.

- 4) I don't think this claim of novelty is correct for these well-known associations:

“We show for the first time that BMI at age 9.5 years and change in BMI over childhood are genetically correlated with a range of cardio-metabolic traits in later life, including adult BMI, glucose-related traits, cholesterol and risk of hypertension, and type 2 diabetes. “

- 5) Wording (in the discussion): “Moderate” in the following sentence is unnecessary confusing as the point here, which I fully support, is that genetic variation has more of an impact on BMI-change in childhood compared to adulthood.

“This indicates that the between individual rate of change in BMI across childhood has a moderate genetic component, whereas rate of change in adulthood is predominantly driven by environmental factors.”

- 6)The GWAS of individual SNPs is a very nice addition and worth highlighting a bit more. Could you elaborate a little bit, in simpler words, on what the identified associations means for our understanding on how the FTO and ADCY3 variants influence BMI development. For example what do we learn from the association of FTO with the quadratic polynomial and the intercept? Did you compare the results with a more traditional GWAS in the same sample, for example BMI at ages near age 1, 8 and 14? And highlight the added benefits for future meta-GWASs with this approach.

Reviewer #4

(Remarks to the Author)

REVIEWER COMMENTS

Reviewer #1 (Remarks to the Author):

Wang and colleagues here describe a new approach to investigate the genetic contribution to child BMI from infancy to late adolescence. They use random regression to utilize the longitudinal design of the ALSPAC cohort that has repeated measures of height and weight in up to 6291 genotyped children. They quantify the SNP-based heritability to between 23-30% across childhood, identify age-varying genetic patterns of BMI by pairwise genetic correlation analyses and perform a comparison with a classical cross-sectional analysis in the same sample. They conclude that there is a different genetic profile operating during infancy compared to later childhood and adolescence.

The results from this novel method are largely consistent with previous studies that also have shown that the genetic influences on BMI changes with time from early to later childhood and to adulthood. As the authors point out – this is important as this implies that studies of genetics of BMI in childhood are likely to discover variants that differ from loci associated with adult BMI.

However, there are also some concerns with the study. While models taking into account the availability of repeated measurements over time have several advantages compared to cross-sectional approaches, they can also be susceptible to model misspecifications, in particular when the number of observations per individual are limited and there are rapid changes in the phenotype and genotype correlations.

The apparently complete disconnection of infant BMI-genetics with later childhood BMI (from 7 years and onwards) seen with their method stands out as the most novel finding and seems at odds with both previous studies and their own cross-sectional analysis in the ALSPAC data. As detailed below, this raises some concerns about the performance of the longitudinal modelling for the earlier timepoints.

There are also some other suggestions that may help improve the paper.

We thank the reviewer for their thoughtful comments and hope that we can appropriately address their concerns.

Main concerns.

1. The model performance at the earlier timepoints seems problematic. Several issues should be addressed

a. The authors suggest (p19) that the genetic contributions to BMI in early childhood (up to age6) appear to be independent of genetic influences in later childhood. This is at odds with previous larger studies that they refer to elsewhere in the manuscript, f ex Alves et al and Helgeland et al that show that there is indeed a lower, but significant shared genetic architecture between BMI at around age 1 and both later childhood and adulthood. It is of course possible that some of this apparent discrepancy could be explained with the

very large CIs of the point estimates, but it is highlighted as one of the key findings of the paper and thus should be scrutinized.

We appreciate the reviewer’s thoughtful observation and hope that we can clarify this point in more detail. The estimated genetic correlations reported in our study and the previous studies (Couto Alves *et al*, Helgeland *et al* and Silventoinen *et al.*) are consistent between BMI at around age one and both later childhood and adulthood, with overlapping 95% confidence intervals on all estimates (see forest plots below). Unfortunately, the standard errors (and subsequently the 95% confidence intervals) on our estimated genetic correlations are large, possibly due to (1) the relatively small sample size available in ALSPAC in comparison to the previous studies and (2) the exclusion of all measurements below age one and limited data between one and 1.7 years. This means our trajectory-based genetic estimates at year one relies only on the relatively small amount of data from age one or older.

Forest plot of the genetic correlation (95% confidence interval) estimates between BMI at age one and BMI at age five from our study (Wang *et al.*) and the previous studies.

Forest plot of the genetic correlation (95% confidence interval) estimates between BMI at age one (adiposity peak for Couto Alves et al) and adult BMI from our study (Wang et al.) and the previous studies.

We have revised the relevant section on page 19, which no longer specifies “independent genetic influences” between early and later childhood. We have also updated the discussion highlighting the consistency between our study and the previous studies, but the larger standard errors from the RRM:

“The low genetic correlations found between BMI in infancy and later childhood, in contrast to the high genetic correlations at subsequent ages, indicate that there are different genetic profiles for BMI across the life-course. Additionally, the SNP-based heritability in early childhood was relatively unaffected by adjusting for an adult BMI PGS, whereas the SNP-based heritability in later childhood attenuated.”

“Our results show decreasing age-to-age genetic correlations as the difference between ages increased from one to 18 years. This is consistent with previous studies including the CODATwins study¹, the MoBa study² and the Early Growth Genetics study³ of BMI at the adiposity peak and adiposity rebound. The genetic correlation estimates from the RRM between one year and from six years onwards were lower than in the previous studies, but the standard errors were large. The large standard errors are likely due to the relatively limited amount of data between one and two years of age in ALSPAC and the exclusion of data prior to age one. These decreasing age-to-age genetic correlations suggest different genetic influences affecting BMI in infancy and early childhood compared to late childhood and adolescence.”

Additionally, we have included the following figure to the supplementary material to show the overlapping confidence intervals around the genetic correlation estimates from the RRM and the MoBa study (Supplementary Figure 10).

Supplementary Figure 10: Comparison of SNP-based genetic correlations between BMI at different ages across early life, estimated using different modelling approaches in ALSPAC (random regression model using longitudinal data) and MoBa (linkage disequilibrium score regression using cross-sectional data).

Each panel shows the genetic correlation between BMI at a given Age target age (title of each subplot) and BMI at other ages. Black lines represent estimates from the random regression model (RRM) in ALSPAC; yellow lines represent estimates from cross-sectional GWAS in MoBa using LD Score Regression (LDSC). Error bars indicate 95% confidence intervals. Note that BMI measurements before age 1 were excluded in ALSPAC but included in the MoBa phenotype definition at year 1, which may contribute to differences in early estimates. Despite differences in data structure and modelling, both approaches show higher genetic correlations between temporally adjacent timepoints.

b. The authors own pairwise correlations between cross-sectional models in Alspac suppl Table 11 shows a picture that is more similar to previous studies with point estimates 0.58 (SE:0.15) and 0.68 (SE:0.15) to age 7 for age 0.8Y and 1.7 Y respectively, and continued relatively strong positive correlation estimates with measures later in life. Again, large SEs can blur the picture, but conclusions from these two analyses seem to be largely contradictory.

We agree with the reviewer that the point estimates for the genetic correlations estimated from cross-sectional analyses using GCTA are more similar to the previous studies, but the standard errors are even larger than those from the RRM. As can be seen in the following figure (which has been added to the supplementary material of the manuscript, Supplementary Figure 11), the 95% confidence intervals overlap at all time points with the 95% confidence intervals around the estimates from the RRM. The developers of GCTA recommend a large sample when performing GREML analyses (at least 3,160 unrelated individuals will give a standard error of around 0.1 on the SNP-heritability estimate according to their website; <https://yanglab.westlake.edu.cn/software/gcta/#FAQ>). Given the relatively small sample size available in ALSPAC for performing these cross-sectional analyses (sample size ranges from 3,373 individuals at age 17.5 years to 5,350 at age 7.6 years), it is not surprising that the standard errors are large. We included these cross-sectional analyses as a benchmark for our RRM estimates.

In addition, we have also included both in the figure below and Supplementary Table 14 (previously Supplementary Table 11) a cross-sectional analysis at 3.7 years (chdb4) for comparison.

Supplementary Figure 11: Comparison of the SNP-based genetic correlation patterns between BMI at different ages across early life in ALSPAC using the random regression model (RRM) and cross-sectional genome-based restricted

maximum likelihood (GREML) using GCTA.

Each panel displays the SNP-based genetic correlation between BMI at a given target age (title of each subplot) and BMI at other ages. Black lines represent estimates from the RRM applied to longitudinal ALSPAC data. Yellow lines represent estimates based on cross-sectional analyses using GCTA. Error bars indicate 95% confidence intervals. ALSPAC lacks dense follow-up data around 1 year of age, and thus, the 0.8-year cross-sectional estimate was used as a proxy.

c. Model fitting at ages 1-5 years of age. If the phenotypic and genetic correlations change drastically from 1 to 6 years of age as most studies suggest, including the current one, any modelling of the growths curves during this time will rely on their ability to capture these changes. Thus, the number and timepoints of measurements are very important. Table 1 and suppl Fig 1 show that there are indeed very few measurements during this time for 90% of the participants. The BMI measurements in Alspac come largely from two different approaches. Alspac has a CIF-subsample of appr 700 children with dense measurements from 12 to 61 months, but for the remainder the main data are collected from around age 0.8Y, 1.7Y, 3.7Y, and then from age 7Y and onwards. As the authors exclude any measurements before 1Y of age, >90% of the samples have only at best measures at 1.7 and 3.7 years to guide their models in this critical time window. Suppl Fig 1 gives a rather clear illustration of this problem. Thus it can be questioned if the results for age1-3 are robust.

We agree that the time between ages one to five years is a critical developmental window, with substantial changes in BMI around both the adiposity peak and the adiposity rebound, and that sparse data in this period can impact model fitting. As the reviewer notes, ~10% of participants are part of the CIF subsample with high-density measurements; however, it is worth highlighting that 4,454 and 4,255 participants (approximately 70% of the cohort) have measurements at 1.7 and 3.7 years, respectively. This means the majority of participants have at least one (5561, 88%), and many have at least two (4108, 65%) data points measured in a clinic setting during this window.

The random regression model is designed to handle unbalanced and irregularly spaced longitudinal data and has been shown to be relatively robust to missingness (Warrington et al. *Stat. Appl. Genet. Mol. Biol.* 2014)⁴. We also refer to prior ALSPAC publications that have successfully modelled BMI trajectories across this age span (Warrington *et al.*, *PLOS One*, 2013⁵, Warrington *et al.*, *IJE* 2015⁶). Additionally, the model uses information from the CIF samples with dense data to inform the likely trajectories of those individuals with sparse data. We believe that Supplementary Figure 1 effectively illustrates this, where the model appears to fit well to the observed BMI measures in those individuals with sparse data from age one to five years. However, without access to BMI measurements within this early time period, it is impossible to know how well the model is fitting during this time window. This indeed is a limitation of using the ALSPAC data for this research, and we have added the following statement outlining this to the manuscript:

“Finally, although ALSPAC is one of the most comprehensive longitudinal datasets available for such analyses, it is not without limitations. ... There is a sparsity of data between ages one and five years, particularly outside the CIF subsample, where the BMI trajectory changes substantially. ... Therefore, future replication of these findings in more diverse cohorts with dense longitudinal data in early childhood and different drop-out mechanisms is warranted.”

d. It looks like the fit at later time points also may be suboptimal as many subjects lack measurements after age 15. However, it is not likely that the genetic architecture is changing as much here, and thus, interpolation of the growth curve would suffer less.

It is correct that fewer participants have repeated BMI measurements in later adolescence due to loss of follow-up. ALSPAC includes three waves of data collection at in this period—Teen Focus 3 (N = 3,885), Teen Focus 4 (N = 3,052), and “Your Son/Daughter at 16+” (N=2,375). We have now quantified that 4,767 (76%) of participants included in the genetic analysis have at least one measurement after age 15, and 3,235 (51%) have at least two. The RRM assumes that data is missing at random and, if this holds, the estimated parameters will not be biased. However, higher BMI has been shown to be associated with lower continued participation rates (i.e. individuals are more likely to drop-out of the study if they have higher BMI) in ALSPAC, indicating that the data (particularly due to drop-out) might not be missing at random and therefore there may be some bias in our estimated parameters^{7,8}. Such non-random drop out is not unique to ALSPAC⁹, but it is currently unknown exactly how it would bias the estimated parameters or how to adjust the RRM to account for such bias. Therefore, we have added the following limitation to the discussion:

“Finally, although ALSPAC is one of the most comprehensive longitudinal datasets available for such analyses, it is not without limitations. ... Additionally, previous studies have shown that ALSPAC participants are generally of higher socio-economic position^{10,11} and drop out from the study is associated with BMI^{7,8}. This indicates that the missing at random assumption of the RRM might not hold. Therefore, future replication of these findings in more diverse cohorts with dense longitudinal data in early childhood and different drop-out mechanisms is warranted.”

2. The abstract could be clearer on the most novel aspects of the study.

We have revised the abstract to fit within the 150 word limit, simplify the technical language and highlight the novel use of random regression modelling to estimate age-varying genetic architecture in longitudinal studies.

“Body mass index (BMI) changes throughout life with age-varying genetic contributions. We used a random regression model to investigate the genetic contribution to BMI trajectories from ages one to 18 years in 6,291 ALSPAC participants with 65,930 repeated BMI measurements. The estimated SNP-based heritability of BMI at 9.5 years was 28.4% (SE=4.8%), and 23.8% (SE=4.2%) for rate of change in BMI from one to 18 years. The genetic correlations between early childhood and adolescence were low (genetic correlation between two and 17 years is 0.108 (SE=0.146)). We found that the first principal component of the trajectory,

explaining 89% of genetic variation, captures effects which increase in magnitude from early childhood to adolescence and then plateau. A second axis had opposite effects on BMI between early and later ages. Our findings demonstrate the value of RRM to reveal age-specific genetic influences on BMI across development.”

3. What is the gain and novelty in the current approach compared to other longitudinal models? There seems to be another approach to longitudinal modelling with the same PI on medRxiv (doi: 10.1101/2024.03.13.24304263). It would be interesting with a discussion on the similarities and differences with this approach as well as other approaches to longitudinal modelling of growth related traits such as the recent study from EGG (<https://doi.org/10.1186/s13059-023-03136-z>)

The longitudinal modelling approaches that the reviewer highlights each answer slightly different research questions. In the current study we are primarily interested in identifying patterns of genetic variation associated with change in BMI by considering the effect of all SNPs simultaneously. For the first time (to our knowledge), we use a genomic-relatedness-matrix based residual maximum likelihood (GREML) analysis, similar to that performed in GCTA, but allowing repeated measures in BMI across time to be incorporated. While this approach is sometimes used in livestock and plant research, it has rarely been used in human genetic studies. It utilizes all data within a cohort to estimate SNP-based heritability of the BMI trajectory. Because it estimates genetic parameters (e.g., heritability, genetic correlations) of the trajectory, rather than specific features of the trajectory, it enables inference at any arbitrary age point along the trajectory to be determined. In contrast, the other longitudinal methods (Burrows *et al.* (2024) now published in *Nat Comms*¹², and Bradfield *et al.* (2024) *Genome Biology*¹³) use derived parameters from linear mixed models to identify individual SNPs associated with a particular time-point or feature of the growth (BMI or height) trajectory. They subsequently use the genome-wide association statistics in a linkage disequilibrium score regression (LDSC) analysis to estimate (the lower bound of the) SNP-based heritability of those time-points/features. The general differences and similarities between GREML and LDSC have been described in detail elsewhere (see Evans *et al.* (2018) *Nat Genet*, [10.1038/s41588-018-0108-x](https://doi.org/10.1038/s41588-018-0108-x)). For the current study, the RRM is a more powerful approach to estimate SNP-based heritability and genetic correlations within a single cohort as it can appropriately model the variances and covariances between each random effect using individual level data. This can be seen by comparing the standard errors for the SNP-based heritability estimates on linear slope, which are only slightly larger from the RRM in 6,291 ALSPAC individuals ($SE(h_{SNP}^2)$ linear slope = 4.2%) across the whole of childhood (1-18 years) in comparison to the much larger study by Burrows *et al.* (2024) with 19,308 individuals ($SE(h_{SNP}^2)$ linear slope 2-3% across the derived slope parameters). Additionally, we have conducted a GWAS of the linear slope parameter and performed LDSC (see response to reviewer 3, questions 1 and 4 for further details), where the $SE(h_{SNP}^2)=7.5%$ is considerably larger than the RRM.

The functions of age used also differs across the longitudinal modelling approaches. In the current study, a cubic polynomial of age is used in the fixed effects part of the RRM and a quadratic polynomial of age in the random additive genetic effects. In contrast, a cubic spline function is used in Burrows *et al.* (2024) and a non-linear

function is used in Bradfield *et al* (2024)¹³. Each of these functions have previously been compared in Warrington *et al* (2013) *PLOS ONE* (doi: journal.pone.0053897)⁵.

We have added the following to the discussion:

“The SNP-based heritability on change in BMI from one to 18 years (\hat{h}_{SNP}^2 linear slope = 23.8%, SE = 4.2%), which has not previously been described, is substantially higher than that for change in BMI in adulthood (\hat{h}_{SNP}^2 linear slope = 1.98%)¹⁴. This indicates that the between individual rate of change in BMI across childhood has a moderate genetic component, whereas rate of change in adulthood is predominantly driven by environmental factors.”

“The strengths of the current study include the utilisation of RRM, a model rarely used in human genetics research, in combination with data from ALSPAC, a comprehensive long-term birth cohort. The use of a continuous function for genetic variance components of the BMI trajectory enables inference of the parameters (e.g. SNP-based heritability or genetic correlation) at any age on the BMI trajectory. It also leveraged the large number of repeated measurements per individual (an average of 8 BMI measures per individual), improving the precision of our estimates over a traditional cross-sectional approach. We also showed how a PCA of the genetic variance-covariance matrix could revealed patterns of genetic variation in BMI that change over time. For the first time, our approach validates the proposed model of childhood BMI by Couto-Alves and colleagues¹³. Similar models, including linear mixed models^{3,6,12} and Super-Imposition by Translation And Rotation (SITAR)¹³, have been previously used to estimate the effects of individual SNPs on features of the growth (BMI or height) trajectory, but modelling all SNPs simultaneously has not previously been considered.”

“A comparison of polynomial and spline functions has previously been⁵ conducted, and splines have been used to successfully identify the association between individual SNPs and BMI trajectories^{6,12}.”

4. The sample size limits the power to detect differences, but it is indeed interesting to note that the SNP-based h2 point estimates are generally higher in the cross-sectional approach in the current study, but also in previous studies, than the longitudinal approach presented here. Hence, a more extensive discussion on pros and cons for longitudinal vs cross-sectional approaches under various scenarios would be interesting. Alspac, with its two different designs (CIF with high density and higher quality measurements vs a sparser coverage of the majority of samples) could serve as inspiration for how to design new studies.

We have provided an updated Supplementary Table 13, where we include the SNP-based heritability estimated derived from the RRM at the time of each cross-sectional follow-up in ALSPAC and MoBa. This supplementary table shows that the 95% confidence intervals around each of the SNP-based heritability estimates overlap at all ages, indicating that the cross-sectional and longitudinal estimates do not differ. However, the SNP-based heritability was estimated to be higher before age 2 in the MoBa cohort, in comparison to ALSPAC (estimated either using the cross-sectional or longitudinal methods). We have included the following in the results:

“We also compared the cross-sectional genetic analyses from the MoBa cohort² to our RRM estimates, where the SNP-based heritability estimates were higher in MoBa than our data before 2 years of age (Supplementary Table 13).”

As discussed in response to the previous question, we believe that the benefit of estimating SNP-based heritability using longitudinal designs is that you can then infer the SNP-based heritability at any age along the trajectory, regardless of whether data has been measured or not. This will allow for easier comparison across cohorts with different follow-up strategies. As large-scale studies increasingly link biobank data with electronic health records (EHR), irregular and sporadic measurement timing will become more common, highlighting the value of flexible models like RRM that can accommodate such complexities and make optimal use of available data.

Supplementary Table 13: Comparison of the SNP-based heritability between BMI at different ages across early life in ALSPAC using the random regression model (RRM), cross-sectional genome-based restricted maximum likelihood (GREML) using GCTA, and MoBa (linkage disequilibrium score regression using cross-sectional data).

Age in RRM	h_{SNP}^2 (95% CI) RRM	Follow-up codes	Mean age (year)	h_{SNP}^2 (95% CI) GCTA	P_gcta ¹	Age in Moba	h_{SNP}^2 (95% CI) Moba	P_moba ¹
1	0.25 (0.13, 0.37)	Child health database 2	0.8	0.28 (0.14, 0.42)	0.75*	1	0.42 (0.34, 0.5)	0.014#
1.5	0.24 (0.12, 0.36)	Child health database 3	1.7	0.26 (0.12, 0.4)	NA	1.5	0.38 (0.32, 0.44)	0.04
2	0.24 (0.14, 0.34)	NA	NA	NA	NA	2	0.33 (0.25, 0.41)	0.14
3	0.23 (0.13, 0.33)	NA	NA	NA	NA	3	0.29 (0.23, 0.35)	0.34
3.7	0.28 (0.18, 0.38)	Child health database 4	3.7	0.17 (0.01, 0.33)	0.47	NA	NA	NA
5	0.25 (0.15, 0.35)	NA	NA	NA	NA	5	0.28 (0.22, 0.34)	0.62
7	0.28 (0.18, 0.38)	NA	NA	NA	NA	7	0.28 (0.2, 0.36)	0.89
8	0.28 (0.18, 0.38)	Focus@7	7.6	0.26 (0.14, 0.38)	0.82	8	0.33 (0.23, 0.43)	0.42
10.7	0.25 (0.17, 0.33)	Focus@10	10.7	0.28 (0.16, 0.4)	0.73	NA	NA	NA
13.9	0.25 (0.17, 0.33)	Teen Focus2	13.9	0.23 (0.09, 0.37)	0.82	NA	NA	NA
15.5	0.27 (0.17, 0.37)	Teen Focus3	15.5	0.26 (0.1, 0.42)	0.90	NA	NA	NA
17.5	0.30 (0.18, 0.42)	Teen Focus4	17.5	0.37 (0.17, 0.57)	0.57	NA	NA	NA

¹ P-value testing whether the estimated h_{SNP}^2 is different from zero.

CI: confidence interval; RRM: random regression model

* Use 0.8-year estimate from GCTA to proxy the one year h^2 . # MoBA includes measurement under year one.

5. Why did they exclude related individuals, mixed models should allow to account for that?

While it is true that mixed linear models can account for relatedness through the genetic relationship matrix (GRM), we excluded related individuals (defined as having a genetic relatedness (π) >0.05) from our analyses to ensure that the SNP-based heritability estimates reflect the heritability attributable to common SNPs, rather than shared environmental factors or familial effects. As shown in Kemper et al. (2021; doi: 10.1038/s41467-021-21283-4)¹⁵, the covariance between different types of relatives (i.e. unrelated vs. close relatives) shows two distinct linear slopes, likely due to changes in linkage disequilibrium, which needs to be modelled independently. Our dataset did not have enough relatives to adequately model genetic variation in close relatives. Therefore, to align our estimates with standard GREML-based approaches and enable comparability across studies, we used unrelated individuals in our analyses.

6. As the results to a large confirm results from other previous larger studies, the most novel aspect of the study might the method. It would be interesting to know whether the random effects model consume a lot of memory and if it might be possible to scala the model to bigger sample sizes and whether it can be extended to enable identification of genetic variants too.

We have now included information on the computational resources required for the random regression modelling. The ASReml job used at least 13.6 GB of the available 32.0 GB of primary workspace, with peak memory utilization reaching 8.36 GB. The model was run using 16 CPUs, with a job running time of 5 hours and 21 minutes and a CPU efficiency of 45.00%. These resource demands indicate that while the model could be computationally intensive with larger sample, it remains feasible for moderate-sized cohorts and can potentially be scaled to larger datasets. Indeed, we have now included a GWAS of the random effects and principal components (eigenfunctions; more details in Reviewer 3, question 1). We have added the following text to the discussion:

“While our findings are broadly consistent with previous studies of BMI genetics across development^{2,16}, the novelty of our study lies in the application of random regression modelling to estimate the genetic covariance structure (Kg) continuously across age. This approach offers flexibility in modelling trajectories using polynomials of age and provides insights into how genetics influence BMI change throughout early life. Random regression models are computationally demanding, particularly for genome-wide analyses in large samples. In our implementation using ASReml, the model required at least 13.6 GB of primary workspace, with peak memory usage at 8.36 GB for the ALSPAC data. Despite this, scalability is feasible. As shown in Kemper et al. (2024)¹⁷, a two-step approach can be employed by direct calculation of individual trait mean and rate of change, and then using a bivariate GREML approach to estimate the genetic (co)variance components. For our GWAS, we used a similar approach to Burrows et al.¹² whereby we used the individual random effects as phenotypes for the intercept, linear slope, quadratic polynomial to identified several known BMI loci. Although we were underpowered to identify novel loci in this study due to the limited sample size, this strategy makes it possible to

extend the random effects modelling framework to larger cohorts and enables future discovery of age-dependent genetic loci influencing BMI.”

Reviewer #2 (Remarks to the Author):

Key results:

The results indicate that there are different genetic profiles that influence BMI between infancy and adolescence. This could be expected by biological developmental stages and previous studies. However, compared to previous reports this uses large number of measurements in a relatively large sample size and full BMI trajectories instead of cross-sectional timepoints plus additional techniques to evaluate age-varying genetic patterns in the BMI development. The weakness maybe a lack of replication in other studies where similar data are available (although some analyses were validated using COADTwins project). The result section is detailed and describes the observations well although it is quite heavy to read. The whole range of different analytical methods were used to illustrate and validate the result and to reach the best outputs from the rich data – excellent work indeed. Figure 2 is interesting, and informative but the details are blurred with a very small text (same concerns Figure 4 as well), so could be improved. Maybe in Figure 3 eigenfunctions could be better or more explicitly explained in the Figure legend. Some parts of the results will be difficult to grasp if someone is not very familiar with the methodology. Overall, as authors acknowledge, increasing our understanding of the mechanisms influencing BMI across early life could lead to strategies to enable early prevention of obesity.

We would like to thank the reviewer for their constructive and encouraging feedback. We have attempted to address your concerns below.

First, we agree that replication is essential, but unfortunately there are no studies similar to ALSPAC in terms of sample size and duration. This is why we performed both internal validation (using cross-sectional models to validate estimates from the RRM) and detailed comparisons with previous studies (with the MoBa study with data from one to eight years and the CODATwins project). We attempted to replicate our findings in 2,245 individuals from the Born in Bradford (BiB) cohort (total number of repeated measures = 8,405) with BMI measures from one to ten years of age. Unfortunately, we experienced convergence issues with the RRM, and when it did converge, the model produced unstable variance estimates. Therefore, we decided this was not informative to incorporate these analyses into the manuscript.

Second, we have revised the results section to make it less heavy and hopefully easier to understand for readers less familiar with random regression and eigenvalue decomposition.

Third, we have made improvements on Figures 2-4. For Figure 2 and Figure 4, we have increased the font size and improved figure resolution to enhance readability. For figure 3, we have added the following text to increase understanding of the eigenfunctions:

“Figure 3: Principal components 1 and 2, representing independent patterns of additive genetic variation on BMI, evaluated as functions of age from one to 18 years.

We conducted a principal component analysis on the genetic covariance matrix and this figure shows eigenfunctions representing the 1st and 2nd principal components (PC1 and PC2). The x-axis denotes age in years, while the y-axis represents the value of the eigenfunctions. The black solid line represents PC1 and explains 89.1% of genetic variance. The dark grey dashed line represents PC2 which captures 8.7% of the genetic variance.”

Validity:

SEE BELOW: The study is highly valid, interesting and well conducted.

Overall, methods and data have been well presented and they seem valid.

However, some methodologies could be explained more explicitly particularly for readers that are not familiar with the approaches (e.g. use of eigenvalues). I also would like to acknowledge a high expertise of the authors of this work.

We agree that some methodological components could benefit from clearer explanation and have revised the methods section accordingly. In particular, we have updated the terminology around the eigen decomposition, and now describe it in terms of principal components, which we believe is more familiar to the genetics audience. The following is the updated methods section describing this:

“We use a principal component analysis (PCA) on genetic covariance matrix, \mathbf{K}_g , to identify independent patterns of genetic variation across the BMI trajectory. Each principal component (PC) captures an orthogonal axis of genetic variation over time – for example, a set of genetic variants that have stable genetic effects on BMI across age. Eigenvalues from the PCA indicate how much of the variance associated with each PC. This decomposition provides insight into how genetic effects change over childhood.”

Significance:

This study is important and interesting because it summarises large amount of previously published information (good introduction + discussion) and shows how study design and approach itself has impact on suggested heritability estimates. Maybe it would be useful to strengthen the text in terms of that genetic correlations are very much dependent on which ages or growth patterns we are comparing. Information is there but it could possibly be spelled out or summarised better. The aspect and role of confounding is, however, discussed in relation to different types of studies (twin-studies, i.e. purely observational studies compared to those that use genetic information as well). The present and previous studies that have aimed to distil that how different genetic factors have impact on different growth stages are important as they reveal also biological pathways that are important for development of body mass. These data could then possibly be used in the future when designing prevention strategies that also integrate genetic information. Overall, we can say that genetic contribution to the development of obesity is substantial and matter for how environmental factors such as nutrition or exercises may impact fat accumulation. The earlier discovery of LEP/LEPR locus which has been associated with BMI specifically across early life and at

the adiposity peak is very important (as noted in the present study, too). The important outcomes from this study were also, as authors state, that genetic effects on BMI are more age specific than environmental effects, and that larger GWASs are needed to finely map the genetic profile of BMI during early stages. This directs the future works.

We have updated the discussion to highlight the age dependent genetic correlations.

“The low genetic correlations found between BMI in infancy and later childhood, in contrast to the high genetic correlations at subsequent ages, indicate that there are different genetic profiles for BMI across the life-course.”

“These decreasing age-to-age genetic correlations suggest different genetic influences affecting BMI in infancy and early childhood compared to late childhood and adolescence.”

Data and methodology:

The team has used the data from the Avon Longitudinal Study of Parents and Children (ALSPAC) cohort. Random regression modelling was used to estimate the genetic covariance matrix of BMI trajectories from ages one to 18 years with 65,930 repeated BMI measurements from 6,291 children. Cut-off age 1 year was used due to the selected modelling option but justification could be better explained (basically this avoids adiposity peak in infancy in the most of the children, but reading the text it shows that measured under one year, ~0'8y, were also used in SNP based heritability cross-sectional analyses – which is fine, i.e. only measurements over 1 y of age were used in main longitudinal analyses). The data are unique and well-documented. The number of children is based on the availability of growth and genetic data which means that less than half of ALSPAC data was used in these analyses. Table 1 also shows the multiple sources of the data which may increase variability or inconsistency? This arises the question that how representative these data are compared to the original full data with same ethnic origin as used here (this information could be added into the supplementary information for example or at least discussed shortly). However, the decrease in the availability of the data is very common and practically unavoidable e.g. due to restricted resources and losses-to-follow up or withdrawal of the consent to use the data (as seemed to happen in ALSPAC as well). In spite of some caveats the study is highly valid, interesting and well conducted. Overall, methods and data have been well presented including inclusion and exclusion criteria that were multiple and explain to some extent the decrease in the sample size. Table 1 generally also shows the complexity in this kind of longitudinal data collections and formation of a final sample for the analyses. The information on genotyping is given at a detailed enough level.

We have attempted to address the data and methodology concerns raised by the reviewer in the following.

First, we provide a clearer justification as to why we excluded data before one year of age in the longitudinal modelling:

“Modelling BMI longitudinally from birth throughout childhood is challenging as the two inflection points (adiposity peak and rebound) make it difficult to find an appropriate function of age that accurately models the steep rise in BMI after birth then the more gradual changes through the adiposity rebound and puberty²⁹. Therefore, we only used data collected between 1 and 18 years in our RRM. Firstly, we excluded data before one year so most individuals will have had their adiposity peak. Secondly, we excluded data after 18 years where BMI plateaus in the majority of individuals.”

We confirm that measurements taken at approximately 0.8 years were used exclusively in cross-sectional SNP-based heritability analyses, as there were no BMI follow-ups precisely at age 1 in ALSPAC. This is stated in the methods:

“First, we estimated SNP-based heritability and genetic correlations in GCTA (version 1.94.1)¹⁹ using selected measurements of BMI at cross-sectional follow-ups (i.e. average ages are 0.8 (closest cross-sectional time point to one year available), 1.7, 3.7, 7.6, 10.7, 13.9, 15.5, and 17.5 years). Each analysis included only one measurement per individual, with sex and age at measurement as covariates. We mostly used the BMI measurements of the same individuals included in the RRM analyses but included some additional records from the Child health database 2 (mean age =0.8 year, N=4799) and Teen Focus 4 (mean age =17 years, N=3373), which were excluded from the RRM as they were outside 1-18-year age range.”

Second, it is indeed unfortunate that less than half of the ALSPAC data were used in these analyses. This is primarily due to the lack of available DNA or consent from the participants or funding for genotyping. We explicitly acknowledge this in the manuscript.

“Finally, although ALSPAC is one of the most comprehensive longitudinal datasets available for such analyses, it is not without limitations. Approximately half of the cohort have provided consent and have been genotyped, limiting the sample size available for genetic analyses.”

Third, there are indeed multiple measurement sources for the BMI data and while measurement from routine health care have previously been shown to be accurate (Howe, Tilling and Lawlor, 2009, doi:[10.1136/adc.2009.162552](https://doi.org/10.1136/adc.2009.162552))¹⁸, parent report of child’s height tends to be overestimated while weight tends to be underestimated (Dubois and Girad, 2007, doi: [10.1093/ije/dyl281](https://doi.org/10.1093/ije/dyl281))¹⁹. We note that the majority of BMI measurements (84.9%) were obtained from clinical assessments (e.g., CIF, CHDB, Teen Focus, and Focus clinics). Including all available data sources helps maximize the sample size and thereby increase statistical power. To account for potential measurement variability across sources, we included the data source as a covariate in all relevant models.

Fourth, as noted by the reviewer, missing data and drop out (loss to follow-up) is very common in large, longitudinal studies such as ALSPAC. We have added a brief summary of this, and the general representativeness of ALSPAC to the British population, to the Discussion to acknowledge this limitation:

“Additionally, previous studies have shown that ALSPAC participants are generally of higher socio-economic position ^{10,11}and drop out from the study is associated with BMI. This indicates that the missing at random assumption of the RRM might not

hold. Therefore, future replication of these findings in more diverse cohorts with dense longitudinal data in early childhood and different drop-out mechanisms is warranted.”

Analytical approach:

The aims of the study and analytical approaches are generally well and clearly presented. As authors present one potential method for investigating genetic effects on traits over time using repeated measures data is the random regression model (RRM). These methods as such are not that novel or newly developed but are appropriate for these purposes because the model offers flexibility in modelling the correlation structure of repeated measurements through the inclusion of random effects for each subject. Authors write that RMM accommodates missing or incomplete data but it is not clear to me how it works in this respect. Does it impute automatically? The “power” of RMM over other methods is described and justified for the present work. Statistical analysis part is details with model descriptions and fitting (this part could be reviewed by specialised biostatistician) but as noted above the readers who are not familiar with all the methods would appreciate further overall description (e.g. in the main document how Eigenvalue decomposition works in spite of that in the supplementary note there is a more detailed description of this). In contrast to previous studies using cross-sectional data, the RMM increases the power to identify patterns of genetic variation by considering the effect of all SNPs simultaneously and it does not attempt to identify individual SNPs associated with a particular time-point or feature of the trajectory. However, these different approaches including time-point or feature of trajectory analyses are valuable and complimentary as they give different information for example in terms of potential preventative approaches (periods during the life-course). This aspect could be acknowledged. This study is also a nice validation and continuum for the study of Couto-Alves et al that was a very first study, as far as we know, to explore genetics of growth patterns and differences between developmental stages related to BMI growth (age-dependent height growth velocity analyses preceded this actually).

We acknowledge that the RRM is not a newly developed method as it has frequently been used in animal genetics. However, we believe this is one of the first applications to human genetic data and is well-suited for modelling longitudinal genetic effects; we have tried to highlight this throughout the manuscript. To further address the reviewers concerns regarding the analytic approach, we provide the following explanations.

The reviewer raises an important point regarding how RRM accommodates missing data. To clarify, RRM does not perform imputation. Rather, missing phenotypic data are accommodated through restricted maximum likelihood estimation under the assumption that data are missing at random. The model estimates genetic and residual variance components using the observed data. This approach allows for the inclusion of individuals with incomplete phenotypic trajectories, maximizing statistical power while avoiding biases associated with imputation. This property makes RRM especially useful for longitudinal datasets with unbalanced or irregular measurement times. We have now clarified this in the Introduction:

“The RRM offers flexibility in modelling the correlation structure of repeated measurements through the inclusion of random effects for each subject and accommodates missing or **irregularly spaced data by leveraging all available repeated measurements per individual. Rather than imputing missing values, the method uses a mixed-effects framework to model population (fixed effects) and individual-specific effects (random effects)** as a continuous function of time (e.g. age), thus estimating the population average trajectory as well as random coefficients to describe each individual’s trajectory. **This allows each participant to contribute to the estimation of longitudinal genetic parameters, even with incomplete follow-up data.**”

We have provided further description on how eigenvalue decomposition works in the methods and results, as outlined in the response to the “Validity” comments from the reviewer.

Finally, we agree with the reviewer that RRM-based approaches are complementary to previously published timepoint-specific or feature-based analyses GWAS that focus on growth features like peak velocity or adiposity rebound. These approaches offer different biological and translational insights. We have added the following to the discussion:

“Similar models, including linear mixed models^{3,6,12} and Super-Imposition by Translation And Rotation (SITAR)¹³, have been previously used to estimate the effects of individual SNPs on features of the growth (BMI or height) trajectory, but modelling all SNPs simultaneously has not previously been considered.”

Suggested improvements:

In the text above I have suggested improvements of clarifications. Overall, this work is nicely presented and valuable. Replication would be an advantage.

We agree with the reviewer that replication would be an advantage. However, as discussed in previous responses to the reviewers, it is difficult to find appropriate cohorts with the breath of data required for such analyses. We have included further text in the discussion relating to this, and specifically state that replication is warranted:

“Therefore, future replication of these findings in more diverse cohorts with dense longitudinal data in early childhood and different drop-out mechanisms is warranted.”

Clarity and context:

Commented already above. Discussion is also well written, informative and includes key aspects to address. This study falls very well into the context of current urgent health matters that we will need to tackle from early childhood onwards.

References:

It looks to me that the main key references are included. Maybe one that examined genetics of PHV between infancy and adolescence would be useful to mention as it is as far as we know the first of its kind where research team tried to understand genetic differences between developmental stages. The paper shows that nicely.

As suggested, we have now cited several studies in the introduction that examine the genetics of PHV, including Cousminer et al. (2013), which was one of the first to investigate genetic influences on PHV using longitudinal data. While these studies focus on height rather than BMI, they illustrate an important early effort to explore developmental-stage-specific genetic effects.

“Only a limited number of studies with smaller sample sizes have modelled growth patterns across time - these have shown promising and complementary findings to cross-sectional analyses^{3,12,13,20}”

Reviewer #3 (Remarks to the Author):

Wang et al. present a well-motivated, well-designed, and thorough manuscript describing the genetic contribution to body mass index (BMI) trajectories through early life. They utilise random regression modelling applied to repeated measures data from the ALSPAC cohort to estimate the SNP-based heritability of, and genetic correlations between, BMI across ages 1-18 years. The ability to model longitudinal data (with missingness) and simultaneously incorporate multiple common SNPs (through a genetic relatedness matrix) empowers the authors to infer the evolving genetic profile of BMI trajectories. They confirm previous findings of the different genetic components of infant, childhood, and adolescent BMI, with adult BMI genetics explaining more of the latter than the former. While methodologically advanced, I think the manuscript lacks new results, so I have made some suggestions below that may improve this.

Major Comments:

1. Page 5, first paragraph: “In contrast to previous studies... and it does not attempt to identify individual SNPs associated with a particular time-point or feature of the trajectory.” Is this meant to signify a strength of the work? I think it is a drawback that the authors cannot identify putatively causal SNPs with this method and should be included in the limitations.

We agree with the reviewer that this was a limitation of our study. Although the original aim of our study was to estimate the SNP-based heritability of BMI across childhood, and the genetic correlations between specific time points in childhood, we can see this perhaps lacks novelty. We have now conducted a genome-wide association analysis of the intercept, linear slope, and quadratic polynomial from the random effects of the RRM and present this in the updated version of the manuscript. We also perform genome-wide association analyses on the first two principal components (eigenfunctions), which are slightly more difficult to interpret, but are presented alongside the other results. However, we note that the sample size is particularly small (typical GWAS published currently have a sample size in the hundreds of thousands, whereas the current sample size in this study is 6,291 individuals) to identify novel loci using this methodology, but we wanted to present the results to show the breadth of analyses that can be performed using the RRM. We have updated the manuscript as follows.

We have updated the introduction to reflect the ability of the RRM to estimate individual SNP effects:

“In contrast to previous studies using cross-sectional data, the random regression model increases power to identify patterns of genetic variation by considering the effect of all SNPs simultaneously **using all available repeated phenotype measurements**. The RRM also reduces the number of parameters estimated from the data compared to previous twin studies which use a series of pairwise comparisons across ages to estimate genetic correlations¹³. It is flexible in that genetic correlations can be estimated at any given pair of ages (within the age range of the data). **Finally, it can also be used to identify individual SNPs associated with a particular time-point or feature of the trajectory.**”

“We estimate the SNP-based heritability and genetic correlations within the age range of one to 18 years, identify patterns of genetic variation in BMI, explore the influence of an adult BMI polygenic score on the genetic variance of childhood BMI, **estimate the effect of individual SNPs on features of the BMI trajectory and estimate the genetic correlation between those features and adult traits and diseases.**”

The GWAS methods and results are described as below:

“Genome-wide association analysis and genetic correlations with adult disease

We fit a reduced RRM in ASReml that included only individual-specific random effects (i.e. removing the random additive genetic effects) for each polynomial term (intercept, linear slope, quadratic polynomial). This was done to reduce potential inflation of GWAS test statistics stemming from the inclusion of a GRM in the derivation of the phenotype. We then used the individual polynomial terms (intercept, slope and quadratic slope) and individual estimates of PC1 and PC2 (i.e. linear combinations of polynomial terms using PC1 and PC2 coefficients as weights) as phenotypes for GWAS. We performed genome-wide association analyses on each of the five derived phenotypes separately using PLINK v2.0, adjusting for the top 10 genetic principal components (PCs) to account for population stratification. Gene names were assigned based on the nearest protein-coding gene using RefSeq annotations via HaploReg v4.2 (<https://pubs.broadinstitute.org/mammals/haploreg/haploreg.php>).

“Genome-wide association analysis and genetic correlations with adult disease

We fitted a reduced RRM, i.e. one excluding the additive genetic effects, to obtain individual (random effect) estimates of important parameters from our model for GWAS. These parameters include individual's intercept, linear and quadratic slope; as well as linear combinations of these parameters representing PC1 and PC2. We conducted a GWAS for these phenotypes and found the genomic inflation for all traits to be low ($\lambda_{GC} = 1.015-1.034$, **Supplementary Figure 8**). We identified three loci associated with intercept ($P < 5 \times 10^{-8}$) in or near *ADCY3* (lead SNP: rs10203482-C, $\beta = 0.018$, $SE = 0.0027$, $P = 6.53 \times 10^{-11}$), *OLFM4* (lead SNP: rs4477562-T, $\beta = 0.023$, $SE = 0.0041$, $P = 3.98 \times 10^{-08}$) and *FTO* (lead SNP: rs55872725-T, $\beta = 0.016$, $SE = 0.0028$, $P = 5.31 \times 10^{-9}$, **Supplementary Figure 9**). In addition, SNPs within the *FTO* locus were associated with linear slope (lead SNP: rs55872725-T, $\beta = 0.0096$, $SE = 0.0012$, $P = 2.92 \times 10^{-15}$), quadratic polynomial (lead SNP: rs9972653-T, $\beta = -0.0046$, $SE = 0.00074$, $P = 4.38 \times 10^{-10}$) and PC1 (lead SNP:

rs55872725-T, $\beta=0.019$, $SE=0.0030$, $P=1.53\times 10^{-10}$, **Supplementary Figure 9**). SNPs within the *ADCY3* locus were associated with PC1 (lead SNP: rs10203482-C, $\beta=0.019$, $SE=0.0029$, $P=3.08\times 10^{-10}$).”

Finally, we summarise the GWAS findings and describe the strengths and limitations of this approach in the discussion:

“Finally, we replicated previous findings that BMI at age 9.5 years (the intercept from our RRM) and change in BMI over childhood are associated with known adult BMI (*FTO* and *ADCY3*)²¹ and childhood obesity/BMI (*OLFM4*)^{6,22} associated loci.”

“For our GWAS, we used a similar approach to Burrows et al.¹² whereby we used the individual random effects as phenotypes for the intercept, linear slope, quadratic polynomial to identified several known BMI loci. Although we were underpowered to identify novel loci in this study due to the limited sample size, this strategy makes it possible to extend our modelling framework to larger cohorts and enables future discovery of age-dependent genetic loci influencing BMI.”

“Finally, we used the random effects estimates for each individual to estimate SNP effects on the features of BMI growth, identifying three known BMI associated loci.”

2. Page 7, last paragraph: It would be useful to know the reasons why outliers were removed (provided by the growthcleanr package), and the number of removals in different data sources. This could help inform future studies that are looking to aggregate data across multiple sources on the reliability of different sources for childhood BMI (for example, clinical vs self-report by a parent).

We agree that understanding the reason for the identified outliers is important, especially for future efforts to aggregate data across cohorts. Outliers are more likely to reflect measurement or data entry errors, or extreme biological cases that may be outside the intended scope of this study.

To address this, we have added the **Supplementary Table 18** to the Supplementary material, which summarises the number of exclusions by reason (based on growthcleanr flags) and by data source (questionnaire or clinic measurement).

Overall, questionnaire-derived measurements made up a smaller proportion (~10%) of the total records, and their exclusion percentages were generally comparable to those from clinic records, where ‘growthcleanr’ retained over 90% of measurements from each source after quality filtering.

3. What are the inflection points in Figure 1B? Do they represent any known phenomena?

While acknowledging that there seems to be fluctuations in SNP-based heritability over childhood, there are large confidence intervals around these estimates and therefore the fluctuations are likely due to sampling variation rather than real change. We are hesitant to interpret these fluctuations and recommend replication in larger cohorts to understand if they represent known phenomena.

4. It would be really interesting to know if eigenfunctions 1 and 2 (or the PGS for these) explain any proportion of variance in adult diseases, such as adult obesity, breast cancer, type 2 diabetes risk, etc. Genetic factors associated with childhood BMI are protective against some adult diseases and risk factors for others – I wonder if your analysis can separate out the effects attributable to different genetic profiles?

We absolutely agree with the reviewer that it would be interesting to investigate whether the observational associations between childhood BMI and adult diseases is partly driven by genetic effects or not. Because the ALSPAC participants are still relatively young and therefore have a low prevalence of the diseases mentioned by the reviewer, we were unable to look at the direct effect of the eigenfunctions on these traits. Likewise, a PGS derived from the GWAS analysis would likely be underpowered due to the low sample size available in ALSPAC. So, instead, we estimated the genetic correlation between each of the features of the childhood BMI trajectory that we performed GWAS on and a range of adult traits and diseases using linkage disequilibrium score regression (LDSC). This gives us an understanding of how similar the genetic underpinnings of these traits are. First, we validate the SNP-based heritability and genetic correlation between the BMI trajectory features estimated in LDSC with that from the RRM. We show that the standard errors on the estimates from LDSC are considerably larger than those from the RRM (as expected), but the 95% confidence intervals overlap. Second, we show that there is moderate to high genetic correlation between both adult BMI and BMI at age eight and our BMI trajectory features (absolute value of r_g ranging from 0.64-1), except eigenfunction 2 (r_g with adult BMI = -0.212 (SE=0.051), r_g with BMI at age 8 = 0.111 (SE=0.125)). Third, we show that the BMI trajectory features are genetically correlated with a range of adult traits, including risk of hypertension, cholesterol levels (ApoA1, HDL, triglycerides), glucose, HbA1c and type 2 diabetes (see new figure). However, the magnitude of these genetic correlations are similar to the genetic correlation between these traits and adult BMI, indicating that adult BMI may be on the causal pathway. Therefore, further analyses are required to investigate this in detail. We have added the following text to the manuscript.

Introduction:

“These individual SNP effects can then be used in downstream analyses such as genetic correlation and Mendelian randomization analyses.”

“We estimate the SNP-based heritability and genetic correlations within the age range of one to 18 years, identify patterns of genetic variation in BMI, explore the influence of an adult BMI polygenic score on the genetic variance of childhood BMI, estimate the effect of individual SNPs on features of the BMI trajectory and estimate the genetic correlation between those features and adult traits and diseases.”

Methods:

“We used linkage disequilibrium score regression (LDSC)^{23,24} to estimate genetic correlations between the GWAS summary statistics for each of the polynomial terms (intercept, linear slope and quadratic polynomial) and PCs with publicly available GWAS summary statistics for a range of outcomes that have previously been linked with childhood BMI. We also estimated the genetic correlation between adult BMI (using summary statistics from Yengo et al 2018²¹) and each of the outcomes to

contrast with our estimates on childhood BMI traits. The list of outcomes and publicly available GWAS summary statistics are reported in Supplementary Table 10.

Results:

“We used LDSC to estimate genetic correlations between the five derived BMI-trajectory phenotypes and a range of published GWAS summary statistics of 26 adult traits (Supplementary Table 10) and one childhood trait for validation. The SNP-based heritability estimates from LDSC were similar to those from the RRM (Supplementary Table 11), despite differences between the models. The LDSC has relatively large standard errors as it uses GWAS summary statistics rather than the individual level data. These large standard errors often resulted in unstable estimates for corresponding genetic correlations. However, we report them to urge others to replicate these findings. We found that the genetic correlations between the intercept, linear slope and quadratic polynomial were similar between LDSC and the RRM (Supplementary Table 12), and moderate genetic correlations between several adult traits and diseases and rate of change in BMI over childhood (linear slope; Figure 6 and Supplementary Table 10); including triglycerides ($r_g=0.317$, 95% CI [0.203, 0.431]), high-density lipoprotein ($r_g=-0.401$, 95% CI [-0.546, -0.256]), apolipoprotein A1 ($r_g=-0.306$, 95% CI [-0.434, -0.178]), HbA1c ($r_g=0.339$, 95% CI [0.210, 0.468]), glucose ($r_g=0.282$, 95% CI [0.147, 0.417]), risk of type 2 diabetes ($r_g=0.578$, SE=0.135) and risk of hypertension ($r_g=0.235$, 95% CI [0.119, 0.351]). The genetic correlations between the top two principal components and selected traits and diseases are presented in **Supplementary Table 10** and **Supplementary Figure 14**. As expected, some adult traits showed opposite genetic correlations with PC1 and PC2 (e.g., HDL, triglycerides, ApoA1, risk of hypertension and risk of type 2 diabetes), consistent with the opposing direction of their genetic effects at age 18 (**Figure 3**).

Discussion:

“This allowed us to investigate the genetic correlation between these features of BMI growth and a range of traits and diseases in adulthood. For the first time, we have shown that BMI growth is associated with a range of cardiometabolic traits; however, given the genetic correlations with adult BMI are of similar magnitude, it is likely that adult BMI is on the causal pathway between childhood growth and these adult traits. Therefore, further investigation into these genetic correlations using causal modelling is warranted.”

5. Page 26, first full paragraph: Could the authors estimate the heritability from RRM at the mean ages used in the cross-sectional GCTA analysis to enable direct comparison? And add a P-value for the difference between the estimates from their RRM and the cross-sectional GCTA estimates.

In response to this question, and to question 4 from reviewer 1, we have estimated SNP-based heritability from the random regression model (RRM) at the mean ages corresponding to the cross-sectional GCTA analyses in ALSPAC (mean ages = 1.7, 3.7, 5, 7, 10.5, 12.7, 13.8, 15.5, and 17.5 years) and MoBa (mean ages = 1, 1.5, 2, 3, 4, 5, 7, and 8 years). The heritability estimates from both approaches are now presented side-by-side in Supplementary Table 13 (presented above in response to Reviewer #1).

To assess the difference between the RRM and GCTA SNP-heritability estimates, we performed Z-tests using the difference in point estimates and their respective standard errors. P-values for these comparisons are now reported in Supplementary Table 13 as well. Overall, the estimates from both approaches are largely consistent across mid-childhood to adolescence, with no statistically significant differences.

We have added the following to the results:

“Additionally, we compared the cross-sectional genetic analyses performed in GCTA on the MoBa cohort ² to our RRM estimates, and the SNP-based heritability estimates were higher in MoBa than ALSPAC before 2 years of age (Supplementary Table 13).”

6. Eigenfunction 2 appears to be distinct between males and females in the CODAtwins, i.e. it decreases monotonically in males but appears to go through a peak during AR and then declines in females. Could the authors check that this is also the case in ALSPAC? What are some possible reasons for this?

We agree that potential sex differences in the genetic effects on BMI trajectories is an important avenue to explore. We conducted sex-stratified analyses in ALSPAC to estimate the genetic (co)variance structure (Kg) separately for males and females. However, most of the variance and covariance terms in the sex-specific Kg matrix had large standard errors (estimate/SE<1.96), reflecting the limited power due to smaller sample sizes when stratified by sex. Therefore, we decided not to proceed with eigen decomposition in the sex-stratified models, as the resulting eigenfunctions would not be reliable. Decomposing a poorly estimated covariance matrix may lead to misleading conclusions, and we felt it would be inappropriate to do so without sufficient precision.

We agree that the sex-specific shape of eigenfunction 2 observed in CODAtwins is intriguing, and future work in larger samples or through meta-analysis of sex-stratified models may help to clarify whether this is a true reflection of the pattern of genetic effects on BMI across childhood or a statistical artifact. We have added a note on this limitation and potential direction to the manuscript discussion.

“This has limited our ability to perform sex-stratified analyses, even though adiposity traits have previously been shown to be sexually dimorphic (for example, Pulit et al. suggest that approximately one-third of all genome-wide associated signals for waist-to-hip ratio are sexually dimorphic.²⁵)”

7. More broadly related to the point (6) above, it might be worth doing all the presented analyses separately in males and females, given the authors have noted that there may be different age-age genetic correlations in males and females in the Introduction (page 4, citation to Silventoinen et al.)

Again, we agree with the reviewer but unfortunately ALSPAC is not large enough to perform such analyses. We have added this as a limitation in the discussion.

“This has limited our ability to perform sex-stratified analyses, even though adiposity traits have previously been shown to be sexually dimorphic (for example, Pulit et al. suggest that approximately one-third of all genome-wide associated signals for waist-to-hip ratio are sexually dimorphic.²⁵)”

Minor Comments:

1. The Introduction is unnecessarily lengthy. For example, the paragraph on adult BMI heritability (page 3, first full paragraph) is not needed given the focus of the manuscript is on childhood BMI.

We agree and have removed the paragraph on adult BMI heritability from the introduction.

2. Page 3, last paragraph: In the sentence “For example, twin studies estimate a heritability of around 40% at age four, presumably where there is a large shared environmental component to BMI”, it is not clear to me what is meant by “where” and “shared environmental component”.

We have changed the sentence to the following:

“Twin studies, for example, estimate the heritability of BMI at age four to be around 40%²⁶, but this increases to around 80% for adolescents (ten to 19 years) ²⁶⁻²⁸ and adults²⁹. ”

3. Page 4, first paragraph: Correlations of between 0.4 to 0.6 are described as “low”. I would argue that these are not low given the range of possible correlation values is only [-1,1] – perhaps use “moderate” instead?

We agree and have changed the term “low” to “moderate” to more accurately reflect the magnitude of correlations between 0.4 and 0.6.

4. Page 14, sentence above equation 13: I don’t understand the term in parentheses “(e.g. eigenvalue 3)” in relation to the rest of the sentence, as it currently seems as though q represent the number of eigenvalues (i.e. 1, 2, or 3) set to 0, rather than just one eigenvalue (i.e. the qth eigenvalue), being set to 0.

We have removed this section of the methods.

5. Page 21, sentence below Equation 16: “... we can see a relatively large [absolute] weight of the intercept...” – Please insert the word absolute as specified.

We have removed this sentence from the results as we have attempted to describe the eigen decomposition in clearer terms for the general reader.

6. Figure 4A and Supp. Figure 6: Can the authors please add confidence intervals around the mean profiles?

We have added confidence intervals around the mean profiles.

7. Page 17, first paragraph of Results: Could the authors speculate on why the individual-specific effects require a cubic term while the population-level effects are quadratic? Has this been observed before in models of childhood BMI?

We have updated the first sentence to show that the fixed effects (population-level) and the individual-specific random effects have a cubic function of age, while the additive genetic random effects have a quadratic function of age.

“Model comparisons indicated that the model with a quadratic **polynomial for the additive genetic component ($k_g = 3$)**, **cubic polynomial for the random individual-specific effects ($k_i = 4$)**, and a **cubic polynomial for each sex in the fixed effects to model the overall population BMI trajectory** was the superior fit to the data (**Supplementary Table 1**).”

8. Supp. Table 7: It would be useful to present the RRM fixed effects from the model un-adjusted for adult BMI PGS side-by-side with the adjusted coefficients to enable comparison.

We have updated Supplementary Table 7 to include the unadjusted fixed effects.

9. Page 26, first full paragraph: Could the authors plot the age-to-age genetic correlation patterns from GCTA cross-sectional analysis, the same way they have done from the RRM analysis, and present them together to enable direct comparison?

We have generated and included plots of the age-to-age genetic correlation patterns from the GCTA cross-sectional analysis, displayed alongside the RRM results, to enable direct comparison (**Supplementary Figures 10 and 11**).

10. Page 26, last line in the first full paragraph: “... which was not different to the RRM..” – Please provide a P-value for the difference.

We have not performed a statistical test comparing the two estimates. We have updated the text to indicate that the 95% confidence intervals overlap between the two estimates.

“For instance, **the 95% confidence interval around** the estimated genetic correlation between Child health database 3 (mean age = 1.7 years) and Focus@7 (mean age = 7.6 years) ($r_g=0.68$, **95% CI = 0.39-0.97**) **overlapped with the 95% confidence interval around** the RRM estimated genetic correlation between 2 and 8 years ($r_g = 0.40$, **95% CI = 0.16-0.64**).”

Reviewer #4 (Remarks to the Author):

We thank the reviewer for their contribution.

References

1. Silventoinen, K. *et al.* Changing genetic architecture of body mass index from infancy to early adulthood: an individual based pooled analysis of 25 twin cohorts. *Int J Obes (Lond)* **46**, 1901-1909 (2022).
2. Helgeland, O. *et al.* Characterization of the genetic architecture of infant and early childhood body mass index. *Nat Metab* **4**, 344-358 (2022).
3. Couto Alves, A. *et al.* GWAS on longitudinal growth traits reveals different genetic factors influencing infant, child, and adult BMI. *Sci Adv* **5**, eaaw3095 (2019).
4. Warrington, N.M. *et al.* Robustness of the linear mixed effects model to error distribution assumptions and the consequences for genome-wide association studies. *Stat Appl Genet Mol Biol* **13**, 567-87 (2014).
5. Warrington, N.M. *et al.* Modelling BMI trajectories in children for genetic association studies. *PLoS One* **8**, e53897 (2013).
6. Warrington, N.M. *et al.* A genome-wide association study of body mass index across early life and childhood. *Int J Epidemiol* **44**, 700-12 (2015).
7. Taylor, A.E. *et al.* Exploring the association of genetic factors with participation in the Avon Longitudinal Study of Parents and Children. *Int J Epidemiol* **47**, 1207-1216 (2018).
8. Cornish, R.P., Macleod, J., Boyd, A. & Tilling, K. Factors associated with participation over time in the Avon Longitudinal Study of Parents and Children: a study using linked education and primary care data. *Int J Epidemiol* **50**, 293-302 (2021).
9. Tyrrell, J. *et al.* Genetic predictors of participation in optional components of UK Biobank. *Nat Commun* **12**, 886 (2021).
10. Boyd, A. *et al.* Cohort Profile: the 'children of the 90s'--the index offspring of the Avon Longitudinal Study of Parents and Children. *Int J Epidemiol* **42**, 111-27 (2013).
11. Fraser, A. *et al.* Cohort Profile: the Avon Longitudinal Study of Parents and Children: ALSPAC mothers cohort. *Int J Epidemiol* **42**, 97-110 (2013).
12. Burrows, K. *et al.* A framework for conducting GWAS using repeated measures data with an application to childhood BMI. *Nat Commun* **15**, 10067 (2024).
13. Bradfield, J.P. *et al.* Trans-ancestral genome-wide association study of longitudinal pubertal height growth and shared heritability with adult health outcomes. *Genome Biol* **25**, 22 (2024).
14. Venkatesh, S.S. *et al.* Characterising the genetic architecture of changes in adiposity during adulthood using electronic health records. *Nat Commun* **15**, 5801 (2024).
15. Kemper, K.E. *et al.* Phenotypic covariance across the entire spectrum of relatedness for 86 billion pairs of individuals. *Nat Commun* **12**, 1050 (2021).
16. Helgeland, O. *et al.* Genome-wide association study reveals dynamic role of genetic variation in infant and early childhood growth. *Nat Commun* **10**, 4448 (2019).
17. Kemper, K.E. *et al.* Genetic influence on within-person longitudinal change in anthropometric traits in the UK Biobank. *Nat Commun* **15**, 3776 (2024).
18. Howe, L.D., Tilling, K. & Lawlor, D.A. Accuracy of height and weight data from child health records. *Arch Dis Child* **94**, 950-4 (2009).
19. Dubois, L. & Girad, M. Accuracy of maternal reports of pre-schoolers' weights and heights as estimates of BMI values. *Int J Epidemiol* **36**, 132-8 (2007).

20. Cousminer, D.L. *et al.* Genome-wide association and longitudinal analyses reveal genetic loci linking pubertal height growth, pubertal timing and childhood adiposity. *Hum Mol Genet* **22**, 2735-47 (2013).
21. Yengo, L. *et al.* Meta-analysis of genome-wide association studies for height and body mass index in approximately 700000 individuals of European ancestry. *Hum Mol Genet* **27**, 3641-3649 (2018).
22. Vogelezang, S. *et al.* Novel loci for childhood body mass index and shared heritability with adult cardiometabolic traits. *PLoS Genet* **16**, e1008718 (2020).
23. Bulik-Sullivan, B.K. *et al.* LD Score regression distinguishes confounding from polygenicity in genome-wide association studies. *Nat Genet* **47**, 291-5 (2015).
24. Bulik-Sullivan, B. *et al.* An atlas of genetic correlations across human diseases and traits. *Nat Genet* **47**, 1236-41 (2015).
25. Pulit, S.L. *et al.* Meta-analysis of genome-wide association studies for body fat distribution in 694 649 individuals of European ancestry. *Hum Mol Genet* **28**, 166-174 (2019).
26. Silventoinen, K. *et al.* Genetic and environmental effects on body mass index from infancy to the onset of adulthood: an individual-based pooled analysis of 45 twin cohorts participating in the COllaborative project of Development of Anthropometrical measures in Twins (CODATwins) study. *Am J Clin Nutr* **104**, 371-9 (2016).
27. Llewellyn, C.H., Trzaskowski, M., Plomin, R. & Wardle, J. From modeling to measurement: developmental trends in genetic influence on adiposity in childhood. *Obesity (Silver Spring)* **22**, 1756-61 (2014).
28. Haworth, C.M. *et al.* Increasing heritability of BMI and stronger associations with the FTO gene over childhood. *Obesity (Silver Spring)* **16**, 2663-8 (2008).
29. Elks, C.E. *et al.* Variability in the heritability of body mass index: a systematic review and meta-regression. *Front Endocrinol (Lausanne)* **3**, 29 (2012).

REVIEWERS' COMMENTS

Reviewer #1 (Remarks to the Author):

I find the manuscript now much improved, and the addition of a GWAS-discovery really shows the applicability of the approach. The study is clearly an important and interesting contribution to the field.

We thank the reviewer for their thoughtful and constructive feedback. Below, we respond point-by-point to the remaining comments and describe the corresponding changes made to the revised manuscript.

I have only a few remaining questions.

1) Regarding the performance of the model during the earlier time-points. I appreciate the thorough answer and the changes to the manuscripts which I think have improved the paper. I still think the authors could help the readers and the design of future studies if they discuss a bit more the implications of their findings for the longitudinal modelling based approaches of BMI: In particular the implications based on the relatively rapid changes in genetic influences/correlations that seem to occur from 1 to 5 years of age and how it may affect the results.

If indeed there is a low genetic correlation, then, it must potentially be a concern when there is a non-random and quite dramatic lack of measurements for the earlier time-points. Based on the sFig1 plots, that shows 16 random individuals, there are at least five individuals (inds 6,7, 9,11,and 14) that do not have a single measurement before the typical adiposity rebound at age 5, and four (2,4,10 and 12) that have only one measurement. I think it would be appropriate to be clearer about this in the limitations and possibilities section – as it might contribute to more noise (and even bias) when modelling these early life genetic influences – and may explain

1. The apparent lack of signal for LEPR and other AP-centred signals, but stronger performance for ADCY3 and FTO.

2. The lower heritability and correlation with later time-points compared to age-stratified approaches (where no modelling is done).

This raises the possibility to speculate a bit on how future studies could be targeted to seek to optimize the trade-off between retaining sample size and avoiding diluting the signals or even introduce bias to the analysis (for ALSPAC the performance at age1-3 will be more dependent on the impact of later time-points – thus AP-centred signals will be diluted). What are the pros and conc for modelling the entire childhood trajectory, rather than splitting it in two: one pre- and one post-AR if they are indeed relatively un-correlated? A bit speculation here would be interesting to read.

We have added the following to the limitations section to make it clearer the sparsity of data in ALSPAC prior to age seven when the regular follow-ups of the full cohort began:

“There is a sparsity of data between ages one and seven years, particularly outside the CIF subsample, where the BMI trajectory changes substantially. For example, of the 6,291 ALSPAC participants included in the analysis, 463 (7%) have no BMI measurements between one and seven years when the regular clinic follow-ups of all participants began, 931 (15%) have one measure and 2209 (35%) have two measures. The sparse data over this time period could produce less precise estimates of the additive genetic and unique individual differences.”

“Therefore, future replication of these findings in more diverse cohorts with dense longitudinal data in early childhood (particularly before the age of seven, where data is sparse in ALSPAC) and different drop-out mechanisms is warranted.”

Regarding the pros and cons for modelling the entire trajectory versus splitting it in two, this choice of modelling depends on the research question of interest, the shape of the trajectory being modelled and the availability of data. The following are a couple examples where splitting the data in two may be justified:

- When the shape of the trajectory clearly displays an abrupt jump in the level or change in slope before and after a specific event. For example, before and after the start of an intervention or puberty onset. Likewise, if there are different mechanisms of action before and after the specific event, i.e. different covariates or variance structures apply to the two time periods. This is not the case for BMI over childhood, where the change is more continuous over time.
- When a single continuous function of time in the model performs poorly (as determined by model diagnostics) or is uninterpretable. This could be the case when a high-order polynomial function over smooths the curve and fails to capture a rapid change. This was indeed seen when we were modelling the BMI trajectory in Sovio et al (2011), where we were unable to capture both the adiposity peak and adiposity rebound using the one continuous function. Hence the data was split into two and the adiposity peak and adiposity rebound were modelled separately. In the current analyses, we decided to exclude data prior to age one, so that we were only modelling one rapid change in BMI, and therefore avoiding any over smoothing (which was also indicated by our model diagnostics).

The data can be split in one of two ways: 1) creating two datasets and fitting a separate model to each one, or 2) fitting a spline function within the one model, which uses knot points to split the data. An added complexity when using either of these methods, is identifying at what age is most appropriate to split the data/add a knot point. This challenge was articulated in Burrows et al (2024)¹, where the first knot point had a large impact on both model fit and estimation of the age at the adiposity peak. These decisions are often not trivial. Additionally, there needs to be enough measurements before and after the split to appropriately model the trajectory. So, in the current analysis, splitting the data before and after the adiposity rebound is unlikely to resolve the issue of sparse data in this time period and produce more precise estimates.

While conscious of the length of the manuscript, we have expanded our strengths section in the discussion briefly to speculate on how best to model longitudinal data in genetic studies more generally:

“The strengths of the current study include the utilisation of RRM, a model rarely used in human genetics research, in combination with data from ALSPAC, a comprehensive long-term birth cohort. While the use of a continuous function for genetic variance components of the BMI trajectory estimates the global genetic effects on the change in BMI over childhood, it also enables inference of the parameters (e.g. SNP-based heritability or genetic correlation) at any age on the BMI trajectory. It also leveraged the large number of repeated measurements per individual (an average of 8 BMI measures per individual), improving the precision of our estimates over a traditional cross-sectional approach. While model diagnostics were appropriate using this continuous function in the ALSPAC data, if systematic deviations are detected then a different modelling approach may be required. For example, the polynomial function may fail to capture the change in slope around the adiposity rebound (i.e. it over-smooths the trajectory), or there might be different variance structures before and after the adiposity rebound, then a more complex spline function^{1,2} or splitting the data into two periods^{3,4} might be worthwhile. However, identifying the most appropriate age to add a knot point for the spline function or split the data is often not trivial.

A few additional comments:

2) Genetic correlations page 7: Please comment, in the following sentence, that these estimates between more distant time-points are surrounded by very huge confidence intervals.

“For example, the genetic correlation between one and two years of age was high ($r_g = 0.948$, $SE = 0.015$), whereas the genetic correlation between one and 10 years was not significantly different from zero ($r_g = -0.009$, $SE = 0.142$).”

We have now clarified in the text that genetic correlation estimates between more distant time-points are surrounded by wider confidence intervals. We note that this pattern of increased uncertainty is not exclusive to the RRM; a similar trend is observed when estimating the genetic correlation using cross-sectional LDSC (see Supplementary Figures 13 and 14).

Revised sentence (page 6):

“For example, the genetic correlation between one and two years of age was high ($r_g = 0.948$, $SE = 0.015$), whereas the genetic correlation between one and 10 years was not significantly different from zero ($r_g = -0.009$, $SE = 0.142$). We note that as the difference between ages increases, the confidence interval around the estimates of genetic correlation also increase, which is also observed when using LDSC (Supplementary Figures 13 and 14).”

3) Fig9 E. Please adapt the y-axis to allow the red line to be seen to avoid misinterpretation of genome wide sign.

We appreciate this suggestion. The y-axis scale of Suppl Figure 11E (previously Suppl Figure 9E) has been adjusted so that the red genome-wide significance line is clearly visible.

4) I don't think this claim of novelty is correct for these well-known associations: "We show for the first time that BMI at age 9.5 years and change in BMI over childhood are genetically correlated with a range of cardio-metabolic traits in later life, including adult BMI, glucose-related traits, cholesterol and risk of hypertension, and type 2 diabetes. "

We agree that genetic correlations between childhood BMI and adult cardio-metabolic traits have been reported previously. However, our analyses extend this literature by quantifying the genetic correlation between these adult traits and the change in BMI slope across childhood (1-18 years). Previous studies have typically examined cross-sectional correlations at discrete ages, but not for the overall rate of change in BMI.

Following the reviewer and editor's guidance, we have toned down the wording and refrained from using words such as "new/novel/first" to acknowledge prior findings while clarifying the distinct contribution of our analysis (page 10):

"We show **for the first time** that BMI at age 9.5 years and change in BMI over childhood are genetically correlated with a range of cardio-metabolic traits in later life, including adult BMI, glucose-related traits, cholesterol and risk of hypertension, and type 2 diabetes. **While previous studies have established correlations between childhood BMI analysed cross-sectionally and cardio-metabolic risks in later life^{5,6}, the present analysis additionally examines genetic correlations with rate of change in BMI across childhood.**"

5) Wording (in the discussion): "Moderate" in the following sentence is unnecessary confusing as the point here, which I fully support, is that genetic variation has more of an impact on BMI-change in childhood compared to adulthood.

"This indicates that the between individual rate of change in BMI across childhood has a moderate genetic component, whereas rate of change in adulthood is predominantly driven by environmental factors."

We have removed the word "moderate" (Discussion, page 11):

"This indicates that the between-individual rate of change in BMI across childhood has a **moderate** genetic component, whereas variation in adulthood is predominantly driven by environmental factors."

6)The GWAS of individual SNPs is a very nice addition and worth highlighting a bit more. Could you elaborate a little bit, in simpler words, on what the identified associations means for our understanding on how the FTO and ADCY3 variants influence BMI development. F ex what do we learn from the association of FTO with

the quadratic polynomial and the intercept? Did you compare the results with a more traditional GWAS in the same sample, f ex BMI at ages near age 1, 8 and 14? And highlight the added benefits for future meta-GWASs with this approach.

While we did not compare our results with GWAS of BMI performed cross-sectionally, we have compared them with previous longitudinal GWAS and candidate gene studies, where our results support the previous findings. We have expanded the discussion (page 10) to clarify the interpretation and implications of the GWAS results:

“The association between variants in *FTO* and the intercept, linear slope, and quadratic terms suggests that *FTO* variants influence mean BMI (around 9.5 years), rate of change, and acceleration of BMI change during childhood, consistent with prior evidence of age-dependent genetic^{2,3}. By contrast, variants in *ADCY3* are associated with only the intercept, implying a more constant influence on BMI across childhood without altering the rate of change, again consistent with previous studies². These findings underscore the value of longitudinal GWAS for capturing dynamic genetic effects over development and suggest potential benefits for future meta-analysis of longitudinal GWAS efforts.”

Reviewer #4 (Remarks to the Author):

We thank the reviewer for their contribution.

1. Burrows, K. *et al.* A framework for conducting GWAS using repeated measures data with an application to childhood BMI. *Nat Commun* **15**, 10067 (2024).
2. Warrington, N.M. *et al.* A genome-wide association study of body mass index across early life and childhood. *Int J Epidemiol* **44**, 700-12 (2015).
3. Sovio, U. *et al.* Association between common variation at the *FTO* locus and changes in body mass index from infancy to late childhood: the complex nature of genetic association through growth and development. *PLoS Genet* **7**, e1001307 (2011).
4. Couto Alves, A. *et al.* GWAS on longitudinal growth traits reveals different genetic factors influencing infant, child, and adult BMI. *Sci Adv* **5**, eaaw3095 (2019).
5. Helgeland, O. *et al.* Characterization of the genetic architecture of infant and early childhood body mass index. *Nat Metab* **4**, 344-358 (2022).
6. Voegelezang, S. *et al.* Novel loci for childhood body mass index and shared heritability with adult cardiometabolic traits. *PLoS Genet* **16**, e1008718 (2020).